METHODS

# Artificial neural networks for model identification and parameter estimation in computational cognitive models

**Milena Rmus[1]\***, **Ti-Fen Pan[1]**, **Liyu Xia[2]**, **Anne G. E. Collins[1,3]**

**1** Department of Psychology, University of California, Berkeley, Berkeley, California, United States of America, **2** Department of Mathematics, University of California, Berkeley, Berkeley, California, United States of America, **3** Helen Wills Neuroscience Institute, University of California, Berkeley, Berkeley, California, United States of America

\* milena_rmus@berkeley.edu

**Data Availability Statement:** The data in this manuscript is simulated using the code stored in the referenced GitHub repository, and can be generated by running the simulation code.

## Abstract

Computational cognitive models have been used extensively to formalize cognitive processes. Model parameters offer a simple way to quantify individual differences in how humans process information. Similarly, model comparison allows researchers to identify which theories, embedded in different models, provide the best accounts of the data. Cognitive modeling uses statistical tools to quantitatively relate models to data that often rely on computing/estimating the likelihood of the data under the model. However, this likelihood is computationally intractable for a substantial number of models. These relevant models may embody reasonable theories of cognition, but are often under-explored due to the limited range of tools available to relate them to data. We contribute to filling this gap in a simple way using artificial neural networks (ANNs) to map data directly onto model identity and parameters, bypassing the likelihood estimation. We test our instantiation of an ANN as a cognitive model fitting tool on classes of cognitive models with strong inter-trial dependencies (such as reinforcement learning models), which offer unique challenges to most methods. We show that we can adequately perform both parameter estimation and model identification using our ANN approach, including for models that cannot be fit using traditional likelihood-based methods. We further discuss our work in the context of the ongoing research leveraging simulation-based approaches to parameter estimation and model identification, and how these approaches broaden the class of cognitive models researchers can quantitatively investigate.

## Author summary

Computational cognitive models occupy an important position in cognitive science research, as they offer a simple way of quantifying cognitive processes (such as how fast someone learns, or how noisy they are in choice selection), and testing which cognitive theories offer a better explanation of the behavior. To relate cognitive models to the behavioral data, researchers rely on statistical tools that require estimating the likelihood

Accessible at: https://github.com/MilenaCCNlab/MI-PEstNets.git.

**Funding:** This work was supported by the National Institutes of Health (NIH R21MH132974 to AGEC). The funders had no role in study design, data collection and analysis, decision to publish, or preparation of the manuscript.

**Competing interests:** The authors have declared that no competing interests exist.

of observed data under the assumptions of the cognitive model. This is, however, not possible to do for all models as some models present significant challenges to likelihood computation. In this work, we use artificial neural networks (ANNs) to bypass likelihood computation and approximation altogether, and demonstrate the success of this approach applied to model parameter estimation and model comparison. The proposed method is a contribution to ongoing development of modeling tools which will enable cognitive researchers to test a broader range of theories of cognition.

This is a *PLOS Computational Biology* Methods paper.

## Introduction

Computational modeling is an important tool for studying behavior, cognition, and neural processes. Computational cognitive models translate scientific theories into algorithms using simple equations with a small number of interpretable parameters to make predictions about the cognitive or neural processes that underlie observable behavioral or neural measures. These models have been widely used to test different theories about cognitive processes that shape behavior and relate to neural mechanisms [1–4]. By specifying model equations, researchers can inject different theoretical assumptions into most models, and simulate synthetic data to make predictions and compare these against observed behavior. Researchers can quantitatively arbitrate between different theories by comparing goodness of fit [5, 6], across different models. Furthermore, by estimating model parameters that quantify underlying cognitive processes, researchers have been able to characterize important individual differences (e.g., developmental: [7–10]; clinical: [11–15]) as well as condition effects [16, 17].

Researchers' ability to benefit from computational modeling crucially depends on the availability of methods for model fitting and comparison. Such tools are available for a large group of cognitive models (such as, for example, reinforcement learning and drift diffusion models). Examples of commonly used model parameter fitting tools include maximum likelihood estimation (MLE, [18]), maximum a-posteriori (MAP, [19]), and sampling approaches ([20, 21]). Examples of model comparison tools include information criteria such as AIC and BIC [5, 22], and Bayesian group level approaches, including protected exceedance probability [23, 24]. These methods all have one important thing in common—they necessitate computing the likelihood of the data conditioned on models and parameters, thus limiting their use to models with tractable likelihood. However, many models do not have a tractable likelihood. This severely limits the types of inferences researchers can make about cognitive processes, as many models with intractable likelihood might offer better theoretical accounts of the observed data. Examples of such models include cases where observed data (e.g. choices) might depend on latent variables—such as the unobserved rules that govern the choices [25–27], or a latent state of engagement (e.g., attentive/distracted, [28, 29]) a participant/agent might be in during the task. In these cases, computing the likelihood of the data often demands integrating over the latent variables (rules/states) across all trials, which grows exponentially and thus is computationally intractable. This highlights an important challenge—computing likelihoods is essential for estimating model parameters, and performing fitness comparison/model identification, and alternative models are less likely to be considered or taken advantage of to a greater extent.

Some existing techniques attempt to bridge this gap. For example, Inverse Binomial Sampling [30], particle filtering [31], and assumed density estimation [32] provide approximate solutions to the Bayesian inference process in the specific cases. Many of these methods, however, require advanced mathematical expertise for effective use and adaptation beyond specific cases they were developed for, making them less accessible many researchers. Approximate Bayesian Computation (ABC, [33–37]) offers a more accessible avenue for estimating parameters in models limited by intractable likelihoods. More widely employed in cognitive modeling, the approach of basic ABC rejection algorithms involves translating trial-level data into summary statistics. Parameter values of the candidate model are then selected based on their ability to produce simulated data that closely aligns with summarized data, guided by some predefined rejection criterion.

While ABC rejection algorithms provide a useful workaround solution, it's important to acknowledge their inherent limitations. Specifically, ABC results are sensitive to the choice of summary statistics (and rejection criteria), and the sample efficiency of ABC scales poorly in cases of high-dimensional data [33, 38, 39]. Recent strides in the field of simulation-based inference/likelihood-free inference have addressed these limitations by using artificial neural network (ANN) structures designed to optimize summary statistics, and consequently infer parameters. These methods enable automated (or semi-automated) construction of summary statistics, minimizing the effect the choice of summary statistics may have on the accuracy of parameter estimation [38, 40–44]. This innovative approach serves to amortize the computational the cost of simulation-based inference, opening new frontiers in terms of scalability and performance [40, 41, 45–50].

Here, we test a related, general approach that leverages advances in artificial neural networks (ANNs) to estimate parameters and perform model identification for models with and without tractable likelihood, entirely bypassing the likelihood estimation (or approximation) step. ANNs have been successfully used to fit intractable models in different fields, including weather models [51] and econometric models [6], and more recently cognitive models of decision making [40, 41]. We develop similar approaches to specifically target the intractability estimation problem in the field of computational cognitive science, including both parameter estimation and model identification, and thoroughly test it in a challenging class of models where there are strong dependencies between trials (e.g., learning experiments).

Our approach relies on the property of ANNs as universal function approximators. The ANN structure we implemented was a recurrent neural network (RNN) with feed-forward layers inspired by [52] (Fig 1) that is trained to estimate model parameters, or identify which model most likely generated the data based on input data sequences simulated by the cognitive model. Our approach is similar to previous work in the domain of simulation-based inference [40, 41], with a difference that such architectures are specifically designed to optimize explicit summary statistics that describe the data patterns (e.g., invertible networks). Here, rather than emphasizing steps involving the reduction of data dimensionality through the creation (and selection) of summary statistic vectors and subsequent inference based on parameter value samples, our focus is on the direct translation of raw data sequences into precise parameter estimates, or the identification of the source model (via implicit summary statistics in network layers).

To validate and benchmark our approach, we first compared it against standard model parameter fitting methods most commonly used by cognitive researchers (MLE, MAP, rejection ABC) in cognitive models from different families (reinforcement learning, Bayesian Inference) with tractable likelihoods. Next, we demonstrated that neural networks can be used for parameter estimation of models with intractable likelihood, and compared it to a standard

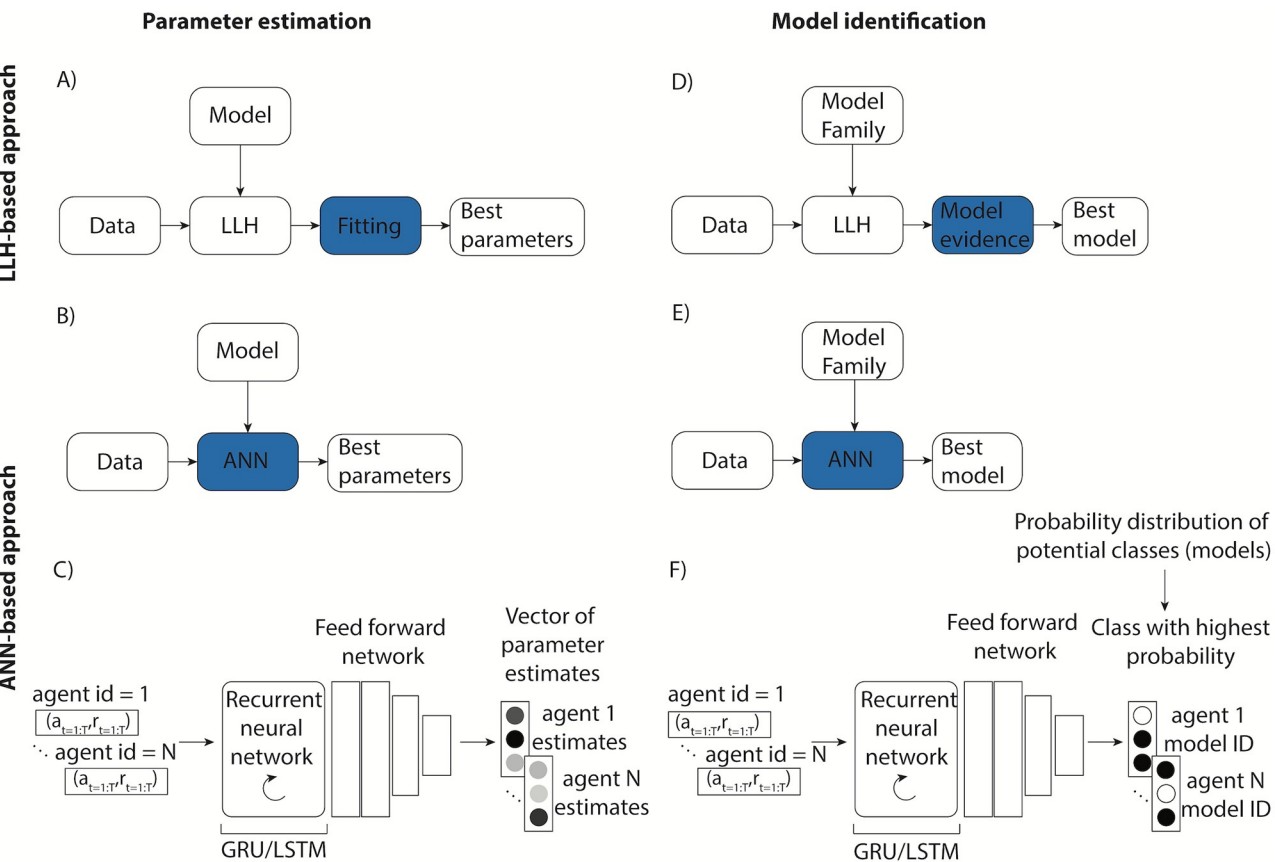

**Fig 1. Artificial neural network (ANN) approach.** A) Traditional methods rely on computing the log-likelihood (LLH) of the data under the given model, and optimizing the likelihood to derive model parameter estimates. B) The ANN is trained to map parameter values onto data sequences using a large simulated data set; the trained network can then be used to estimate cognitive model parameters based on new data without the need to compute or approximate likelihood. C) The ANN structure inspired by [52] is suitable for data with strong inter-trial dependencies: it consists of an RNN and a fully connected feed-forward network, with an output containing ANN estimates of parameter values the data was simulated from for each agent. D) As in parameter estimation, traditional tools for model identification rely on likelihood to derive model comparison metrics (e.g., AIC, BIC) that are used to determine which model likely generated the data. E) ANN, instead, is trained to learn the mapping between data sequences and respective cognitive models the data was simulated from. F) The structure of the ANN follows the structure introduced for parameter estimation, with the key difference of the final layer containing the probability distribution over classes representing model candidates, with the highest probability class corresponding to the model the network identified as the one that likely generated the agent's data.

approximation method (ABC). Finally, we showed that our approach can also be used for model identification.

Our results showed that our method is highly successful and robust at parameter and model identification while remaining technically lightweight and accessible. We highlight the fact that our method can be applied to standard cognitive data sets (i.e., with an arbitrarily small number of participants, and a normal number of trials per participant), as the ANN training is fully done on a large simulated data set. Our work contributes to the ongoing research focusing on leveraging artificial neural networks to advance the field of computational modeling, and provides multiple new avenues for maximizing the utility of computational cognitive models.

## Results

We focused on two distinct artificial neural network (ANN) applications in cognitive modeling: parameter estimation and model identification. Specifically, we built a network with a

structure suitable for sequential data/data with time dependencies (e.g. recurrent neural network (RNN); [52]). Training deep ANNs requires large training data sets. We generated such a data set at minimal cost by simulating a cognitive computational model on a cognitive task a large number of times. Model behavior in the cognitive task (e.g. a few hundred trials of stimulus-action pairs or stimulus-action-outcome triplets (depending on the task) for each simulated agent) constituted ANN's training input; true known parameter values (or identity of the model) from which the data was simulated constituted ANNs' training targets. We evaluated the network's training performance in predicting parameter values/identity of the model in a separate validation set, and tested the trained network on a held out test set. We tested RNN variants and compared their accuracy against traditional likelihood-based model fitting/identification methods using both likelihood-tractable and likelihood-intractable cognitive models. See Methods section for details on the ANN training and testing process.

## Parameter recovery

**Benchmark comparison.**    First, we sought to validate our ANN method and compare its performance to existing methods by testing it on standard likelihood-tractable cognitive models of different levels of complexity in the same task: 2-parameter ($2P - RL$) and 4-parameter ($4P - RL$) reinforcement learning models commonly used to model behavior on reversal tasks [7, 14, 53, 54], as well as Bayesian Inference model ($BI$) and Bayesian Inference with Stickiness ($S - BI$) as an alternative model family that has been found to outperform RL in some cases [55–57]. We estimated model parameters using multiple traditional methods for computing (maximum likelihood and maximum a-posteriori estimation; MLE and MAP) and approximating (Approximate Bayesian Computation; ABC) likelihood. We used the results of these tools as a benchmark for evaluating the neural network approach. Next, we estimated parameters of these models using two variants of RNNs: with gated recurrent units (GRUs) or Long-Short-Term-Memory units (LSTM).

We used the same held-out data set to evaluate all methods (the test set the ANN has not observed yet, see simulation details). For each of the methods, we extracted the best-fit parameters, and then quantitatively estimated the method's performance as the mean squared error (MSE) between estimated and true parameters across all agents. Methods with lower MSE indicated better relative performance. All of the parameters were scaled for the purpose of loss computation, to ensure comparable contribution to loss across different parameters. To quantify overall loss for a cognitive model, we averaged across all individual parameter MSE scores; to calculate fitting method's MSE score for a class of cognitive models (e.g. likelihood tractable models) we averaged across respective method's MSE scores for those models (see Methods for details about method evaluation).

First, we examined the performance of standard model-fitting tools (MLE, MAP and ABC). The standard tools yielded a pattern of results that are expected based on noisy, realistic-size data sets (with several-hundred trials per agent). Specifically, we found that MAP outperformed MLE (Fig 2A, average MSEs: $MLE = .67$, $MAP = .35$), since the parameter prior applied in MAP regularizes the fitting process. ABC was also worse compared to MAP (Fig 2A, average MSE: $ABC = .53$). While the fitting process is also regularized in ABC, worse performance in some models can be attributed to signal loss that arises from approximation to the likelihood. Next, we focused on the ANN performance; our results showed that for each of the models, ANN performed better than or just as well as the traditional methods (Fig 2A, average MSEs for different RNN variants: $GRU = .32$, $LSTM = .35$). Better network performance was more evident for parameter estimation in more complex models (e.g., models with a higher number of

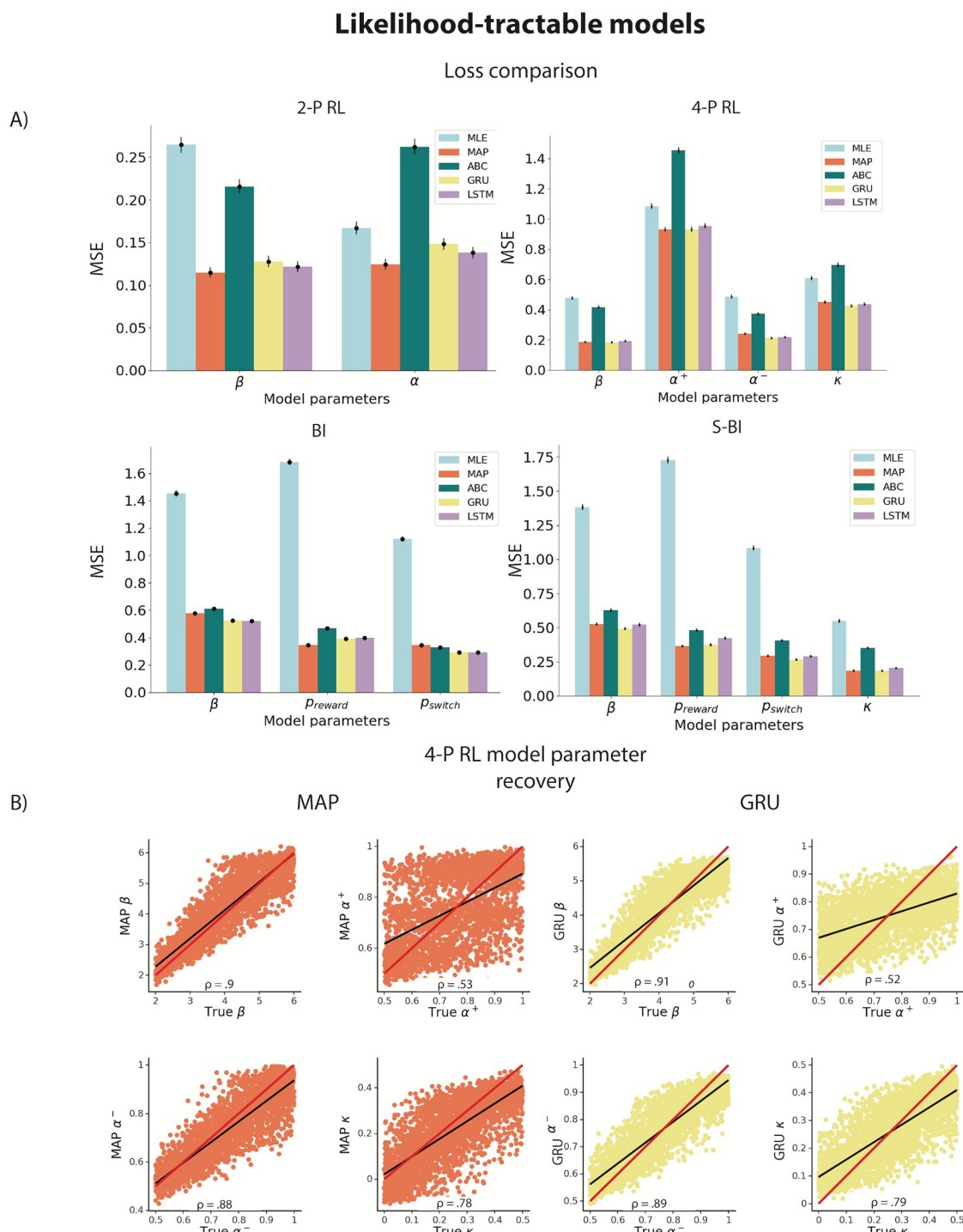

**Fig 2.** A) Parameter recovery loss from the held-out test set for the tractable-likelihood models (2P-RL, 4P-RL, BI, S-BI) using each of the tested methods. Loss is quantified as the mean squared error (MSE) based on the discrepancy between true and estimated parameters. Bars represent loss average for each parameter across all agents, with error bars representing standard error across agents. B) Parameter recovery from the 4P-RL model using MAP and GRU. $\rho$ values represent the Spearman $\rho$ correlation between true and estimated parameters. Red line represents a unity line ($x = y$) and black line represents a least squares regression line. All correlations were significant at $p < .001$.

parameters such as 4P-RL and S-BI; average MSE across these 2 models: *MLE* =.95, *MAP* =.43, *ABC* =.71, *GRU* =.38, *LSTM* =.44).

Next, we visualized parameter recovery. We found that for each of the cognitive models the parameter recovery was largely successful (Spearman $\rho$ correlations between true parameter values and estimated values: $\beta$ $\rho_{MAP}$, $\rho_{GRU}$ = [.90, .91], $\alpha^+\rho_{MAP}$, $\rho_{GRU}$ = [.53, .52], $\alpha^-\rho_{MAP}$, $\rho_{GRU}$ = [.88, .89], $\kappa$: $\rho_{MAP}$, $\rho_{GRU}$ = [.78, .79], Fig 2B; all correlations were significant at $p <.001$). For conciseness, we only show recovery of the more complex model parameters from the RL model family (and only MAP method as it performed better compared to ABC and MLE, as well as only GRU since it performed better than LSTM), as we would expect a more complex model to emphasize superiority of a fitting method more clearly compared to simpler models. Recovery plots of the remaining models (and respective fitting methods) can be found in S2–S5 Figs. Our results suggest that 1) ANN performed as well as traditional methods in parameter estimation based on the MSE loss; 2) more complex models may limit the accuracy of parameter estimation in traditional methods that neural networks appear to be more robust against. We note that for the $4P-RL$ model, parameter recovery was noisy for all methods, with some parameters being less recoverable than others (e.g. $\alpha^+$, Fig 2B). This is an expected property of cognitive models applied to realistic-sized experimental data as found in most human experiments (i.e., a few hundred trials per participant). To check whether the limited recovery can be attributed to parameter identifiability rather than pitfalls of any specific method, we looked at the correlation between parameter estimates obtained using the standard model fitting method (MAP) and the ANN (GRU) (S10 Fig)—with parameters that are not well recovered (e.g., $\alpha^+$ in 4P-RL model) being of particular interest. High correlation between estimated parameters obtained via 2 methods imply systematic errors in parameter identification that apply to both methods—thus suggesting that the weaker correlation between true and fit parameters for some parameters is more likely due to limitations in the model applied to the data set than method specifications such as poor optimization performance. We further discuss the implications in the discussion section—highlighting that computational models should be carefully crafted and specified regardless of the tools used for model fitting.

**Testing in cognitive models with intractable likelihood.** Next, we tested our method in two examples of computational models with intractable likelihood. As a comparison method, we implemented Approximate Bayesian Computation (ABC), alongside our ANN approach to estimate parameters. The two example likelihood-intractable models we used had in common the presence of a latent state which conditioned sequential updates: RL with latent attentive state ($RL-LAS$), and a form of non-temporal hierarchical reinforcement learning ($HRL$, [27]). Since we cannot fit these models using MAP or MLE we used only ABC as a benchmark. Because we found LSTM RNN to be more challenging to train and achieve similar results when compared to GRU, we focused on GRU for the remainder of comparisons. We found that average MSE was much lower for the neural network compared to ABC for both RL-LAS (Fig 3A, average MSEs: *ABC* =.62, *GRU* =.21) and HRL (Fig 3A, average MSEs: *ABC* =.28, *GRU* =.19). Spearman correlations were noisier for ABC compared to GRU in both models (Fig 3B, **RL-LAS:** $\beta$ $\rho_{ABC}$, $\rho_{GRU}$ = [.72, .91], $\alpha$ $\rho_{ABC}$, $\rho_{GRU}$ = [.83, .95], T $\rho_{ABC}$, $\rho_{GRU}$ = [.5, .81]; **HRL:** $\beta$ $\rho_{ABC}$, $\rho_{GRU}$ = [.86, .89], $\alpha$ $\rho_{ABC}$, $\rho_{GRU}$ = [.85, .9]; all correlations were significant at $p <.001$). Furthermore, some parameters were less recoverable than others (e.g., the T parameter in RL-LAS model, which indexed how long participants remained in an inattentive state); this might be in part due to less straightforward effect of T on behavior (S6 Fig). Note that, in order to obtain our ABC results we had to perform an extensive exploration procedure to select summary statistics—ensuring reasonable ABC results. Indeed, the choice of summary statistics is not trivial and represents an important difficulty of applying basic rejection ABC [33, 38], that we can entirely bypass using our new neural network approach. We acknowledge that recent

## Likelihood-intractable models

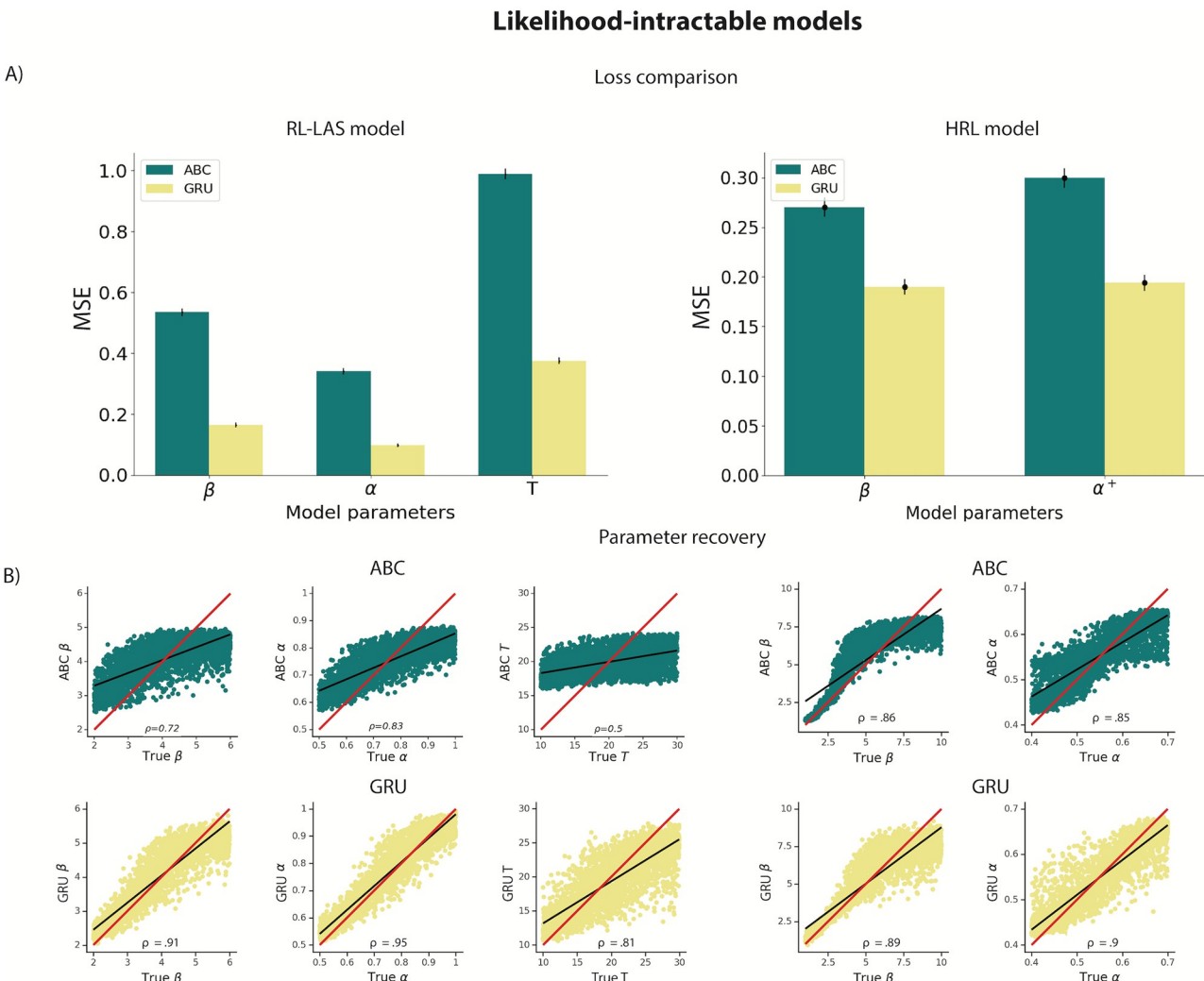

**Fig 3.** A) Parameter recovery loss from the held-out test set for the intractable-likelihood models (RL-LAS, HRL) using ABC and GRU network. Loss is quantified as the mean squared error (MSE) based on the discrepancy between true and estimated parameters. Bars represent MSE average for each parameter across all agents, with error bars representing standard error across agents ((S17 Fig) shows variability across seeds). B) Parameter recovery from the RL-LAS and HRL models using ABC (green) and GRU network (yellow). $\rho$ values represent the Spearman $\rho$ correlation between true and estimated parameters. Red line represents a unity line ($x = y$) and black line represents a least squares regression line. All correlations were significant at $p < .001$.

methods that rely on ANNs replaced standard ABC methods by automating (or semi-automating) the construction of summary statistics [38, 40–44, 51]. However, we aimed to explore an alternative approach, independent of explicit optimization of summary statistics, and focused on the ABC instantiation that has been most frequently implemented in the field of cognitive science as a benchmark [33–35].

**Uncertainty of parameter estimates.** Thus far, we have outlined a method that provides point estimates of parameters based on input data sequences, as is typically the use for much lightweight cognitive modeling (e.g., maximum likelihood estimation or MAP). However, it is sometimes also valuable to compute the uncertainty associated with these estimates [21]. It is possible to extend our approach to add this capability. While there are various alternative ways to do so (e.g. Bayesian neural networks), the approach we have opted for is incorporating

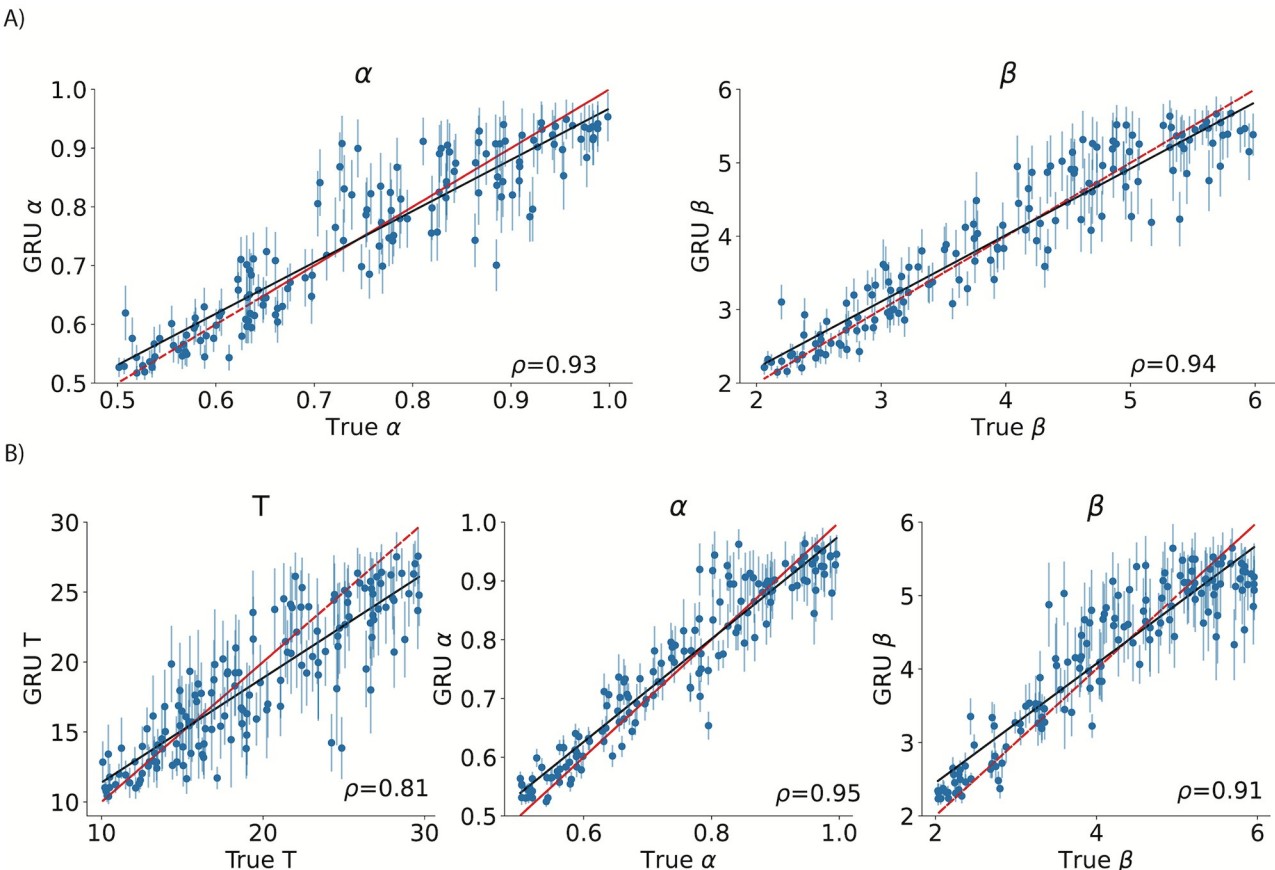

**Fig 4.** Using evidential learning to evaluate uncertainty of parameter estimates for A) the 2-parameter RL model (tractable likelihood) and B) the RL model with latent attention states (intractable likelihood). Vertical lines around point estimates illustrate model uncertainty. We are showing only 100 data points for cleaner visualization, Spearman $\rho$ values are computed based on the total number of agents in the held-out test data (3k).

evidential learning into our method [58]. Evidential learning differs from Bayesian networks in that it places priors over likelihood function, rather than network weights. The network leverages this property to learn both statistical (aleatoric) and systematic (epistemic) uncertainty during the process of estimating a continuous target based on the input data sequences. This marks a shift from optimizing a network to minimize errors based on average prediction, without considering uncertainty.

We applied our method with integrated evidential learning to tractable and intractable versions of the RL models (2P-RL and RL-LAS, Fig 4). We found that incorporating this modification did not compromise the point estimate parameter recovery (e.g. compared to our baseline method focused only on maximizing the accuracy of the predictions). Additionally, it enabled the estimation of the uncertainty around the point estimate, as demonstrated by [58]. This extension appears to be more computationally expensive (with longer training periods) than our original method, but not to a prohibitive extent.

## Model identification

We also tested the use of our ANN approach for model identification. Specifically, we simulated data from different cognitive models, and trained the network to make a prediction regarding which model most likely generated the data out of all model candidates. The

network architecture was identical to the network used for parameter estimation, except that the last layer became a classification layer (with one output unit per model category) instead of a regression layer (with one output unit per target parameter).

For models with tractable likelihood, we performed the same model identification process using AIC [5] that relies on likelihood computation, penalized by the number of parameters, to quantify model fitness as a benchmark. We note that another common criterion, BIC [6], performed more poorly than AIC in our case. The best fitting model is identified based on the lowest AIC score—a successful model recovery would indicate that the true model has the lowest AIC score compared to other models fit to that data. To construct the confusion matrix, we computed best AIC score proportions for all models, across all agents, for data sets simulated from each cognitive model (Fig 5; see Methods).

As shown in Fig 5A, model identification performed using our ANN approach was better compared to the AIC confusion matrix, with less "confusion"—lower off-diagonal proportions compared to diagonal proportions (correct identification). Model identification using AIC is likely in part less successful due to some models being nested in others (e.g. $2P - RL$ in $4P - RL$, $BI$ in $S - BI$). Specifically, since AIC score represents a combination of likelihood and the penalty incurred by the number of parameters it is possible that the data from more complex models is incorrectly identified as better fit by a simpler version of that model (e.g. the model with fewer parameters; an issue which would be more pronounced if we used a BIC confusion matrix). The same phenomenon is observed with the network, but to a much lesser extent, showing better identification out of sample—even for nested models. Furthermore, the higher degree of ANN misclassification observed for $BI/S - BI$ was driven by $S - BI$ simulations with a stickiness parameter close to 0, which would render the $BI$ and $S - BI$ non-distinguishable (S7 Fig).

Because we cannot compute the likelihood for our likelihood-intractable models based on closed-form solutions via MAP, we only report the confusion matrices obtained from our ANN approach. In the first confusion matrix, we performed model identification for $2P - RL$ and $RL - LAS$, as we reasoned these two models differ by only one mechanism (occasional inattentive state), and thus could potentially pose the biggest challenge to model identification. In the second confusion matrix, we included all models used to simulate data on the HRL task (*HRL model*, *Bayesian inference model*, *Bayesian inference with stickiness model*). In both cases, the network successfully identified the correct models as true models, with a very small degree of misidentification, mostly in the nested models. Based on our benchmark comparison to AIC, and the proof of concept identification for likelihood intractable models, our results indicate that the ANN can be leveraged as a valuable tool in model identification.

## Robustness tests

**Robustness tests: Influence of different input trial sequence lengths.**   ANNs are sometimes known to fail catastrophically when data is different from the training distribution in minor ways [59–62]. Thus, we investigated the robustness of our method to differences in data format we might expect in empirical data, such as different numbers of trials across participants. Specifically, we conducted robustness experiments by varying the number of trials in each individual simulation contributing to training or test sets, fixing the number of agents in the training set.

To evaluate the quality of parameter recovery, we used the coefficient of determination score ($R^2$), which normalizes different parameter ranges. We found that the ANNs trained with a higher trial number reach high $R^2$ scores in long test trials. However, their performance suffers significantly with a smaller number of test trials. The results also show a similar trend

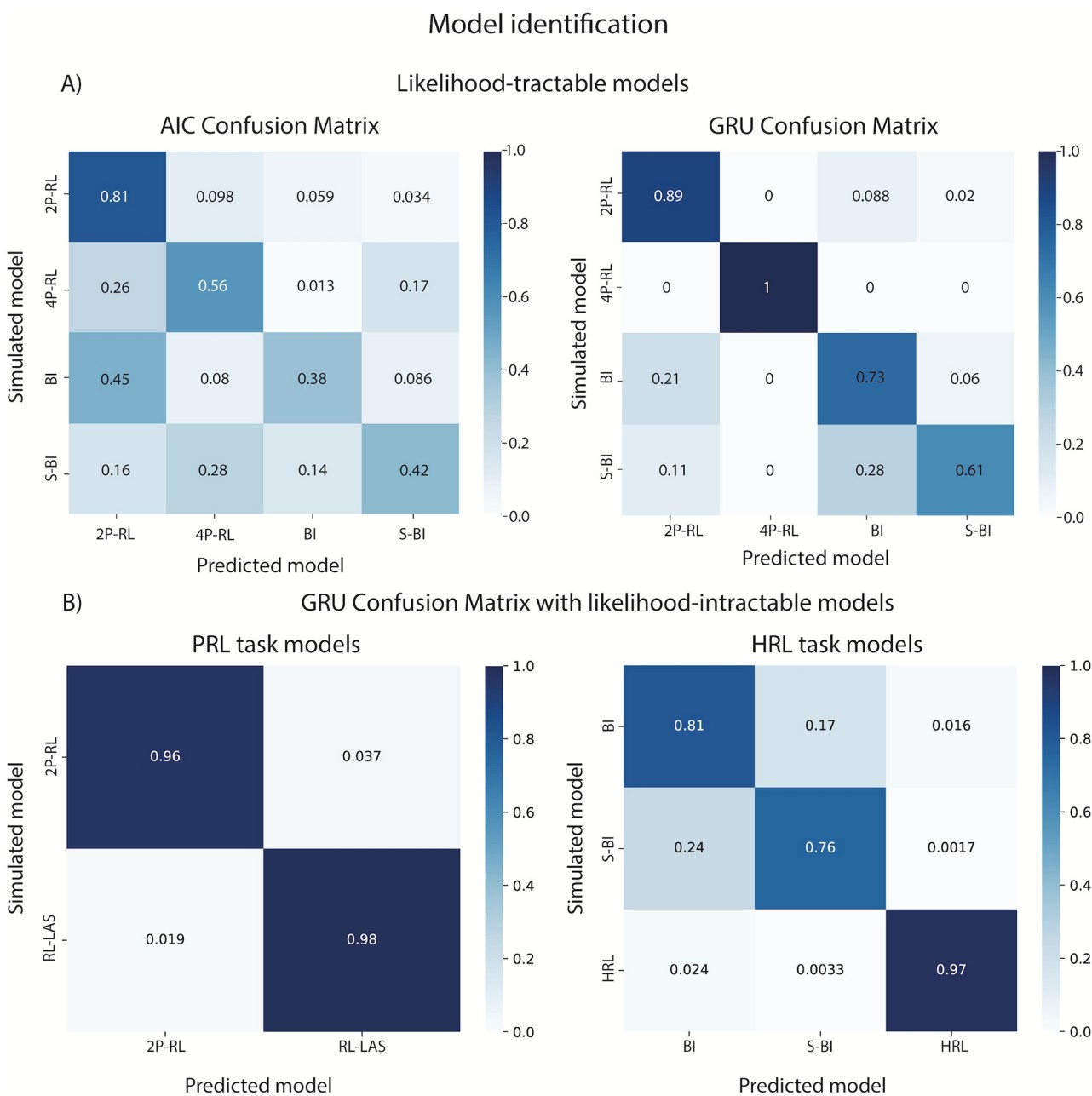

**Fig 5. Model identification results.** A) Confusion matrix of likelihood-tractable models from the PRL task based on 1) likelihood/AIC metric, and 2) ANN identification. AIC confusion matrix revealed a much higher degree of misclassification (e.g., true simulated model being incorrectly identified as a different model). B) Confusion matrix of likelihood-intractable models using ANN (2P-RL and RL-LAS models were simulated on the PRL task; HRL, BI and S-BI models were simulated on the HRL task).

in model identification tasks except, that training with higher trial number doesn't guarantee better performance. For instance, the classification accuracy between HRL task models of the ANN trained with 300 trials reaches 87% while the ANN trained with 500 trials is 84%.

Data-augmentation practices in machine learning increase robustness of models during training [63] by introducing different types of variability in the training data set (e.g., adding noise, different data sizes). Specifically, slicing time-series data into sub-series is a data-

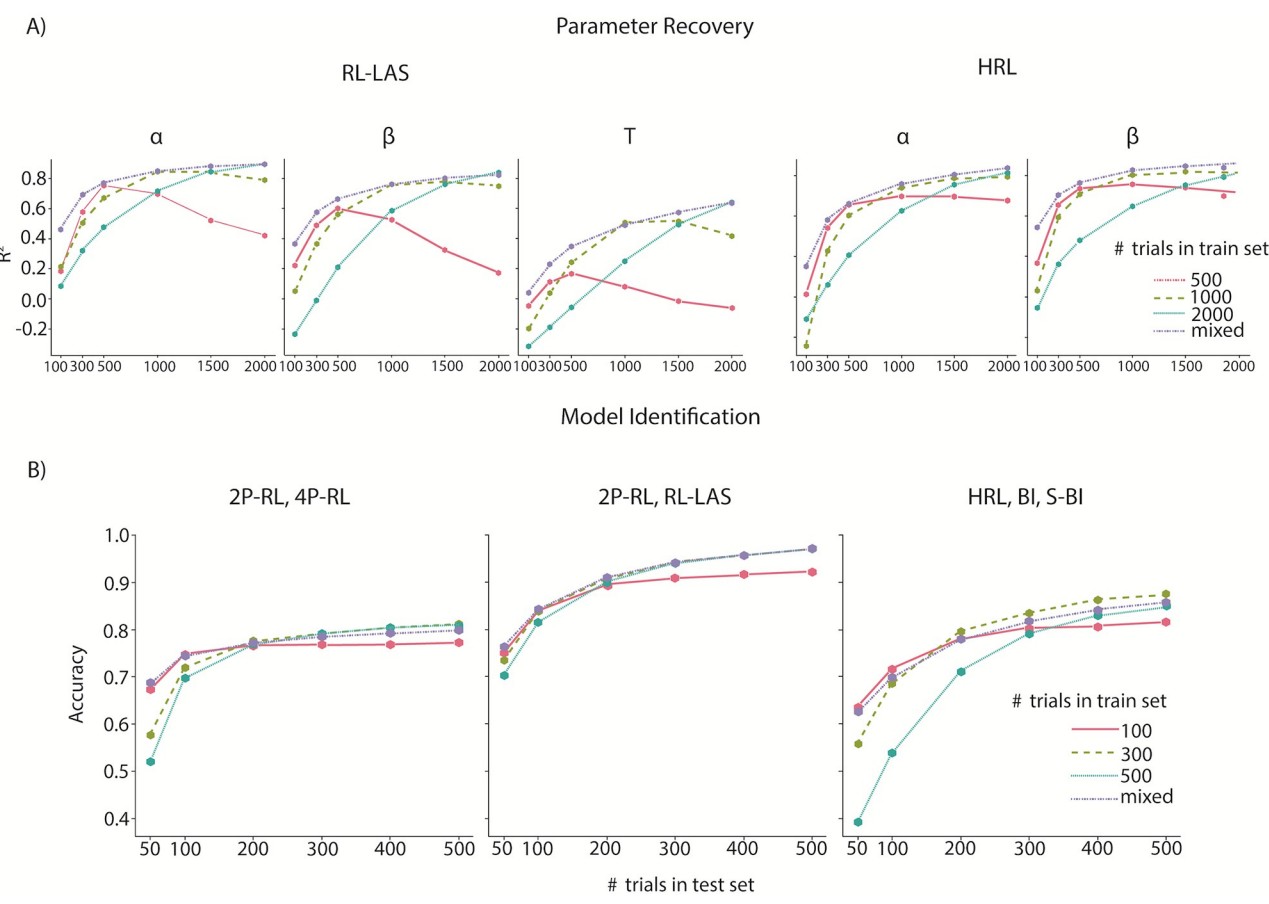

**Fig 6. Robustness checks using different training (different line colors) and testing (x-axis) trial sequence lengths.** A) Parameter estimation in both RL-LAS and HRL models shows that training with a mixture of trial sequence lengths (purple line) yields more robust out-of-sample parameter value prediction compared to fixed trial sequence lengths. B) Best model identification results, performed on different combinations of model candidates, were also yielded by mixed trial sequence length training. The number of agents/simulations used for training was kept constant across all the tests (N agents = 30k).

augmentation practice that increases accuracy [64]. Thus, we trained our ANN with the fixed number of simulations with different trial numbers. As predicted, we found that the ANNs trained with a mixture of trial sequence lengths across simulations (purple line) consistently yielded better performance across different numbers of test trials for both parameter recovery and model identification (Fig 6A and 6B).

**Robustness tests: Prior parameter assumptions.**   We also tested the effects of incorrect prior assumptions about the parameter range on method performance. Specifically we 1) trained the network using data simulated from a narrow range of parameters (theoretically informed) and 2) trained the network based on broader range of parameter values. Next, we tested both networks in making out-of-sample predictions for test data sets that were simulated from narrow and broad parameter ranges, respectively. The network trained using a narrow parameter range made large errors at estimating parameters for data simulated outside of the range it was trained on; training the network on a broader range overall resulted in smaller error, with some loss of precision for the parameter values in the range of most interest (e.g., the narrow range of parameters the alternative network is trained on). We observed similar results with MAP, where we specified narrow/broad prior (where narrow prior would place

high density on a specific parameter range). Notably, training the network using a broader range of parameters while oversampling from a range of interest yielded more accurate parameter estimation compared to MAP with broad priors (Approach described in S9 Fig).

**Robustness tests: Model misspecification.**   In addition to testing the effects of incorrect priors, we also tested the effect of model misspecification on standard method and ANN performance (focusing on MAP and GRU network, as they performed the best in parameter recovery tests on benchmark models). We fit the Bayesian inference model (without stickiness) to the data simulated from the Bayesian inference model with stickiness using MAP. For the ANN, we trained the neural network to estimate parameters of the Bayesian inference model, and tested it on the separate test set data simulated from the Bayesian inference model with stickiness. For each method, we looked at the correlation between the ground truth Bayesian inference with stickiness model parameters, and the method's parameter estimates (S13 Fig). Our results suggest that the parameters shared between the 2 models are reasonably recoverable using both MAP and ANN (e.g., the recovery is noisier but comparable to that of parameters in Bayesian models without model misspecification (S4 and S5 Figs); furthermore, the correlation between ground truth and estimated values is similar for the two methods.

To make the model misspecification more extreme, we additionally simulated data from a Bayesian inference model, and estimated RL model parameters from the simulated data. We did this using standard methods (MAP) and ANN, and repeated the same process in reverse (simulating data from an RL model, and fitting Bayesian inference model parameters). We found that both MAP and ANN exhibited similar patterns. That is, in the case of simulating the Bayesian inference model and fitting RL model parameters, the estimated $\beta$ captured the variance from the true $\beta$ and $p_{switch}$, while the estimated $\alpha$ parameter captured the variance driven by the Bayesian updating parameters $p_{reward}$ and $p_{switch}$ (S14 Fig). In the case of simulating RL model and fitting Bayesian inference model parameters, the $p_{switch}$ parameter captured the noise in the simulated data coming from the $\beta$ parameter, and the variance from the $\alpha$ parameter was attributed to the $p_{reward}$ parameter (S15 Fig). We also correlated parameter estimates generated by the two methods. High correlation implies that MAP and GRU generate similar parameter estimates, suggesting that they are impacted by model misspecification in a similar way (S11 Fig).

## Discussion

Our results demonstrate that artificial neural networks (ANNs) can be successfully and efficiently used to estimate best fitting free parameters of likelihood-intractable cognitive models, in a way that is independent of likelihood approximation. ANNs also show remarkable promise in successfully arbitrating between competing cognitive models. While our method leverages "big data" techniques, it does not require large experimental data sets: indeed, the large training set used to train the ANNs is obtained purely through efficient and fast model simulation. Thus, our method is applicable to any standard cognitive data set with a normal number of participants and trials per participants. Furthermore, while our method requires some ability to work with ANNs, it does not require any advanced mathematical skills, making it largely accessible to the broad computational cognitive modeling community.

Our method adds to a family of approaches from other attempts at using neural networks for fitting computational cognitive models. Specifically, previous work leveraging amortized inference has focused on taking advantage of large-scale simulations and invertible networks. This approach involves training the summary segment of the network to adeptly learn relevant summary statistic vectors, while concurrently training the inference segment of the network to approximate the posterior distribution of model parameters based on the outputs generated by

the summary network [40, 41, 46]. This method has successfully been applied to both parameter estimation and model identification (and performs in a similar range as our method when applied to intractable models we implemented in this paper), bypassing many issues of ABC. In parallel, work by [47] showcased Likelihood Approximation Networks (LANs) as a method that approximates likelihood of sequential sampling models (but requires ABC-like approaches for training), and recovers posterior parameter distributions with high accuracy for a specific class of models (e.g., drift diffusion models); more recently, [48] used a similar approach with higher training data efficiency. Work by [65] used Approximate Bayesian Computation (ABC) in conjunction with mixture density networks to map data to parameter posterior distributions. Unlike most of these approaches our architecture is not dependent on [47, 48] or explicitly designed to optimize [40, 41, 46] summary statistics. By necessity, hidden layers of our network do implicitly compute a form of summary statistic that are translated into estimated parameters/model class in the output layer; however, we do not optimize for such statistics explicitly, beyond their ability to support parameter/model recovery.

Other approaches have used ANNs for different purposes than fitting cognitive models [66]. For example, [52] leveraged the flexibility of RNNs (which inspired our network design) to map data sequences onto separable latent dimensions that have different effects on decision-making behavior of agents, as an alternative to cognitive models that make more restrictive assumptions. Similarly, work by [67] also used RNNs to estimate RL parameters and make predictions about behavior of RL agents. Our work goes further than this approach in that it focuses on both parameter recovery and model identification of models with intractable likelihood, without relying on likelihood approximation. Furthermore, multiple recent papers [68, 69] have used ANNs as a replacement for cognitive models, rather than as a tool for supporting cognitive modeling as we do, demonstrating the number of different ways ANNs are taking a place in computational cognitive science.

It is important to note that while ANNs may prove to be a useful tool for cognitive modeling, one should not expect that their use immediately fixes or overrides all issues that may arise in parameter estimation and model identification. For instance, we have observed that while ANNs outperformed many of the traditional likelihood-based methods, recovery for some model parameters was still noisy (e.g., learning rate $\alpha$ in the 4P-RL model, Fig 2). This is a property of cognitive models when applied to experimental applied to data sets that range in hundreds of trials. Standard methods (e.g., MAP) fail in a similar way—as shown by the high correlation between MAP and ANN parameter estimates (S10 Fig), which suggests that parameter recovery issues have more to do with identifiability limitations of the data and model, rather than other issues such as optimization method. Similarly, model parameters are often not meaningful in certain numerical ranges, and sometimes model parameters trade off in how they impact behavior through mathematical equations that define the models—making the parameter recovery more challenging. Furthermore, when it comes to model identification, particularly with nested models, the specific parameter ranges can influence the outcome of model identification, favoring simpler models over more complex ones (or vice versa). This was evident in our observations regarding the confusion between Bayesian inference models with and without stickiness, wherein the ground truth values of stickiness played a decisive role in the model identification. This is to say ANNs should be treated as a useful tool that is only useful if the researchers apply significant forethought to developing appropriate, identifiable cognitive models.

In a similar vein, it is important to recognize that the potential negative implications of model misspecification extend to neural networks, much like they impact traditional model-fitting approaches. For instance, our estimation of parameters may be conducted under the assumption of model X, whereas, in reality, model Y might be the most suitable for explaining

the data—leading to poor parameter estimation and model predictions. Our test of the systematic effects of model misspecification involved utilizing a network trained to estimate parameters from one model (e.g. Bayesian Inference) to predict parameters for the held-out test set data simulated from a different model (e.g. Bayesian Inference with stickiness, or RL). We compared this to model misspecification with a standard MAP approach. Notably, neither method exhibited significant adverse effects. When models were nested, the parameters shared between the two models were reasonably well recovered. When the model misspecificpation was more extreme (with models from different families), we again observed similar effects on the two methods, where variance driven by one parameter tended to be recovered similarly. Thus, our approach appears equally (but not worse) subject to the risk of model misspecification as other fitting methods. In light of these findings, our key takeaway is to exercise caution against assuming that the use of a neural network remedies all issues typically associated with modeling. Instead, we advocate for the application of conventional diagnostics (e.g., model comparison, predictive checks) that are commonly employed in standard methods to ensure robust and accurate results.

Relatedly, we have shown that the parameter estimation accuracy varies greatly as a function of the parameter range the network was trained on, along with whether the underlying parameter distribution of the held out test-set is included in that range or not. This is an expected property of ANNs that are known to underperform when the test data systematically differs from training examples [59–61]. As such, the range of parameters/models used for inputs constitutes a form of prior that constrains the fit, and it is important to carefully specify it with informed priors (as is done with other methods, such as MAP). We found that training the network using a broader parameter range, while heavily sampling from a range of interest (e.g., plausible parameter values based on previous research) affords both accurate prediction for data generated outside of the main expected range, with limited loss of precision within the range of interest (S9 Fig). This kind of practice is also consistent with practices in computational cognitive modeling, where a researcher might specify (e.g., using a prior) that a parameter might range between two values, with most falling within a certain, more narrow range.

One factor that is specific to ANN-based methods (as opposed to standard methods) is the effect different hyperparameters (e.g., size of the neural network, choice of the learning rate, dropout values, etc.) may have on network performance—commonly resulting in overfitting or underfitting. We observed that the network performance, particularly in parameter recovery, is most significantly influenced by the number of units in the GRU layer and the chosen dropout rate. A suitable range for the number of GRU units is typically between 90 and 256, covering the needs of most cognitive models. A dropout rate within the range of 0.1 to 0.2 is generally sufficient. We have outlined the details of parameter ranges we tested in the table (S1 Table). To address this challenge, we employed an automated hyperparameter tuning approach, as outlined by Bergstra, Yamins, and Cox (2013). This Bayesian optimization for tuning hyper-parameters helps reduce the time required to obtain an optimal parameter set by learning from previous iterations. Additionally, in the process of training a neural network, the initialized random weights play a significant role in determining the network's convergence and the final performance. Different random seeds can result in different initializations of the network weights, which may affect the optimization process downstream, and potentially yield different final solutions. It is important to be mindful of this; we have inspected the effects of setting different seeds on our network performance (S17 Fig), and found that overall network performance was stable across different seeds, with slight variations (1 seed) for both parameter estimation and model identification—showcasing the need for cautious practice of inspecting network's performance under multiple seeds.

We compared our artificial neural network approach against existing methods that are commonly used to estimate parameters of likelihood-intractable models (e.g. ABC, [33, 70]). While traditional rejection ABC provides a workaround solution, it also imposes certain constraints. Specifically, it is more suitable for data with no sequential dependencies, and the accuracy of parameter recovery is largely contingent on the selection of appropriate summary statistics, which is not always a straightforward problem. More recent advances in the domain of simulation-based inference [38, 40, 42, 44] solve many ABC-issues by automating the process of construction of summary statistics. For the purpose of this project, we have focused on the methods that are most commonly used in cognitive modeling (e.g. maximum likelihood/ maximum a posteriori), but future work should extend to conducting the same benchmarking procedure involving these inference methods.

Alternative approximation methods (e.g. particle filtering [31]; density estimation [32]); inverse binomial sampling [30] may prove to be more robust, but frequently require more advanced mathematical knowledge and model case-based adaptations, or are more computationally expensive; indeed, some of them may not be usable or tractable in our type of data and models where there are sequential dependencies between trials [30, 71]. ANN-based methods such as ours or others' [40, 41, 49], on the other hand, offers a more straightforward and time-efficient path to both parameter estimation and model identification. Developing more accessible and robust methods is critical for advances in computational modeling and cognitive science, and the rising popularity of deep learning puts neural networks forward as useful tools for this purpose. Our method also offers an advantage of requiring very little computational power. The aim of the project at its current state was not to optimize our ANN training in terms of time and computing resources; nevertheless, we used Nvidia V100 GPUs with 25 GB memory and required at most 1 hour for model training and predictions. This makes the ANN tool useful, as it requires a low amount of computing resources and can be done fast and inexpensively. All of our code is shared on GitHub.

We primarily focused on extensive tests using synthetic data, in particular in the context of learning experiments that present important challenges for some methods (such as BADS [71] or ABC [33–35]) due to the dependency between trials, and have not been thoroughly investigated with other ANN-based approaches. A critical next step will be to further validate our approach using empirical data (e.g., participant data from the tasks). Similarly, we relied on RNNs due to their flexibility and capacity to handle sequential data. However, it will be important to explore different architectures, such as transformers [72], for potentially improved accuracy in parameter recovery/model identification, as well as alternative uses in cognitive modeling.

In addition, our baseline approach lacks the capability to quantify the complete uncertainty in parameter estimation, offering only point estimates. This is similar to many lightweight cognitive modeling approaches (such as MAP and LLH), but stands in contrast to other methods that integrate simulation-based inference with neural network structures [40, 41, 45, 47, 48], where the ability to capture full uncertainty represents a notable strength. Nevertheless, we have showcased that our method can easily be extended to provide uncertainty estimates by incorporating evidential learning techniques [58], at a slight computational cost, but minimal impact on point estimates' accuracy. Furthermore, we included both RL and Bayesian inference models to demonstrate our approach can work with different classes of computational models. Future work will include additional models (e.g. sequential decision making models) to further test robustness of our approach.

In conclusion, we propose an accessible ANN-based method to perform parameter and model identification across a broad class of computational cognitive models for which application of existing methods is challenging. Our work should contribute to a growing literature

focused on developing new methods that will allow researchers to quantitatively test a broader family of theories than previously possible.

## Materials and methods

### Tasks

**Probabilistic reversal learning task.**   We have simulated data from different models (see the Models section) on a simple probabilistic reversal learning task (PRL; [73, 74]). In the task, an agent chooses between two actions on each trial, and receives a binary outcome ($r = 1$ [reward] or $r = 0$ [no reward]). One of the two actions is correct for a number of trials; a correct action is defined as the action that gets rewarded with higher probability (e.g., $p(r = 1 | action = correct) = 0.80$), with $1 - p$ probability of getting no reward if selected. After a certain number of trials, the correct action reverses; thus the action that was previously rewarded with low probability becomes the more frequently rewarded one (S1 Fig). This simple task (and its variants) has been extensively used to provide mechanistic insights into learning from reinforcement, inferring probabilistic structure of the environment, and people's ability (or failure) to update the representation of a correct choice.

**Hierarchical reinforcement learning task.**   We developed a novel task environment that can be solved using a simple but plausible model with intractable likelihood. In this task, an agent observes N arrows (in different colors), each pointing at either left or right direction. The agent needs to learn which arrow is the correct one, by selecting an action that corresponds to either left or right side (consistent with the direction the arrow is pointing at) in order to get rewarded. Selecting the side the correct arrow is pointing at rewards the agent with high probability ($p = .9$); choosing an action by following direction of other arrows leads to no reward ($r = 0$) with same high probability. The correct arrow changes unpredictably in the task, which means that the agent must keep track of which arrow most reliably leads to the reward, and update accordingly when the change occurs. We refer to this task structure as hierarchical because the choice policy (left/right) depends on the higher-level rule (color) agents choose to follow.

### Cognitive models

**PRL task models.**   We implemented multiple models of the PRL task to test the artificial neural network (ANN) approach to parameter estimation. First, we cover the benchmark models; these are the models that we can fit using traditional methods (MLE, MAP), as well as the ANN, to ensure that we can justify using the ANN if it performs at least just as well as (or better than) the traditional methods.

**Reinforcement learning models family.   Two-parameter reinforcement learning model**. We simulated artificial data on the PRL task using a simple 2-parameter reinforcement learning model (2P-RL). The model assumes that the agent tracks the value of each action contingent on the reward history, and uses these values to inform the action selection on each trial.

The model uses simple delta rule to update action values on each trial upon outcome observation, by first computing the reward prediction error (RPE, $\delta$) as the discrepancy between the expected and the observed outcome, and then adjusting the value of the chosen action using the RPE scaled by the learning rate ($\alpha$) [75]:

$$\delta = r - Q_t(a)$$
$$Q_{t+1}(a) = Q_t(a) + \alpha \delta \tag{1}$$

We also allowed for counterfactual updating, where the value of the non-chosen action also gets updated on each trial [7, 76]:

$$\delta_{\text{unchosen}} = (1 - r) - Q_t(1 - a)$$
$$Q_{t+1}(1 - a) = Q_t(1 - a) + \alpha\,\delta_{\text{unchosen}} \tag{2}$$

The action values are transformed into action probabilities using the softmax function, thus defining a policy where actions with higher value are chosen with higher probabilities. The $\beta$ parameter controls how deterministic the choices are with higher values of $\beta$ corresponding to more deterministic choices:

$$P(a) = \frac{\exp(\beta\,Q_t(a))}{\sum_{i=1}^{n_A} \exp(\beta\,Q_t(a_i))} \tag{3}$$

The 2p-RL model contained following free parameters: learning rate ($\alpha$) and softmax beta ($\beta$).

**Four-parameter reinforcement learning model.** The four parameter RL (4P-RL) model follows the same updating and policy structure as the 2-parameter RL, with 2 main differences. The 4P-RL model differentiates between positive and negative feedback ([77]), by using different learning rates—$\alpha^+$ and $\alpha^-$ for updating action values after positive and negative outcomes respectively:

$$Q_{t+1}(a) = \begin{cases} Q_t(a) + \alpha^+ \delta & \text{if } \delta > 0 \\ Q_t(a) + \alpha^- \delta & \text{if } \delta \leq 0 \end{cases}$$

Furthermore, 4P-RL model also includes the stickiness parameter $\kappa$ which captures the tendency to repeat choice from the previous trial:

$$P(a) \propto exp(\beta Q + \kappa\ \text{same}(a, a_{t-1})) \tag{4}$$

Like in the 2P-RL we also included counterfactual updating of values for non-selected actions. The 4P-RL model included following free parameters: positive learning rate ($\alpha^+$), negative learning rate ($\alpha^-$), softmax beta ($\beta$) and stickiness ($\kappa$).

**Bayesian models family. Bayesian inference model**. Bayesian inference model (BI) assumes that an agent infers the latent state in the environment, updates the latent state based on new observations, and uses the inference process to make rewarding choices. For instance, in the PRL task, the agent infers a latent state corresponding to the correct action ($C_t$: $a_{right}$ = $cor$ or $C_t$: $a_{left}$ = $cor$) at time $t$. The agent tracks and updates their belief over which one of the two actions is currently the correct one based on 1) their estimate of the switch frequency ($p_{switch}$) and 2) how noisy the reward is ($p_{reward}$) from the history of observations up to the previous trial $H_{t-1}$. On each trial, the belief is updated according to the Bayes' rule—based on the prior belief (agent's model of the task) and likelihood of observed evidence (the outcome given the choice):

$$p(C_t = i | r_t, a_t, H_{1:t-1}) = \frac{P(r_t | C_t = i, a_t) P(C_t = i | H_{1:t-1})}{\sum j (P(r_t | C_t = j, a_t) P(C_t = j | H_{1:t-1}))}$$

where i and j are in [left/right], $p(C_t = i | H_{1:t-1})$ is the prior probability, and $p(r_t | C_t = i, a_t)$ is the likelihood of outcome given the action. The likelihood is defined in accordance to whether the choice matches the latent state:

$$p(r_t = 1 | a_t = i, C_t = i) = p_{reward}$$

where $p_{reward}$ is the parameter controlling the probability of receiving the reward given the choice of correct action. Posterior belief for the correct action is updated to a prior belief for the upcoming trial in accordance with the $p_{switch}$ parameter, which determines the probability that the correct action might have reversed on the current trial:

$$p(C_{t+1} = i|H_{1:t-1}) = (1 - p_{switch})p(C_t = i|H_{1:t-1}) + p_{switch}(1 - p(C_t = i|H_{1:t-1}))$$

Like in the RL models, the action selection in Bayesian models also followed the softmax policy; however, instead of being informed by the Q values the action probabilities were determined by the belief $W$ given the choice and reward history $H$ and the choice parameter $\beta$:

$$W_{t+1} = p(C_{t+1} = i|H_{1:t}) \tag{9}$$

$$P(a_{t+1}) = \frac{\exp(\beta\ W_i(t+1))}{\sum_{i=j}\exp(\beta\ W_j(t+1))}$$

The BI model included following parameters: inferred probability of reward given the action determined by the current belief ($p_{reward}$), likelihood of the correct action reversing ($p_{switch}$) and softmax beta ($\beta$).

**Bayesian inference model with stickiness.** We also added a variation of the Bayesian inference model that accounts for sticky choice behavior (e.g., repeating actions) by introducing a stickiness parameter $\kappa$ that augments the belief associated with the action chosen on the previous trial:

$$W_{t+1} = p(C_{t+1} = i|H_{1:t}) + \kappa(i = a_t)$$

**Intractable likelihood.** As a proof of concept, we implemented a simple model that assumes a latent state of agent's attention (engaged/disengaged). This model can't be fit using methods that rely on computing likelihood. While models can have intractable likelihood for a variety of reasons, we focused on leveraging latent variables (e.g., attention state), that are not readily observable in the data. Thus, in the data that is being modeled, only the choices are observed—but not the state the agent was in while executing the choices. The learned choice value which affects the choice likelihood depends on the trial history, including which state the agent was in. Thus, if there are 2 such states, there are $2^N$ possible sequences that may result in different choice value estimates after N trials. To estimate choice values and likelihood on any given trial one must integrate over the uncertainty of an exponentially increasing latent variable—thus making the likelihood intractable.

**RL and latent engagement state.** We simulated a version of a 2p-RL model for a probabilistic reversal learning (PRL) task that also assumes that an agent might occupy two of the latent attention states—engaged or disengaged— during the task (RL-LAS). The model assumes that in the engaged state an agent behaves in accordance with the task structure (e.g., tracks and updates values of actions, and uses action values to inform action selection). In the disengaged state, an agent behaves in a noisy way, in that 1) it does not update the Q value of actions, and 2) chooses between the two actions randomly (e.g., uniform policy) instead of based on their value (e.g., through softmax). Note that assumption 1) is different from a previous version of the model our group considered [78, 79], and is the core assumption that renders the likelihood intractable. The agent can shift between different engagement states at any point throughout the task, and the transition between the states is controlled by a parameter $\tau$. Specifically, for each agent we initialized a random value $T$ between 10 and 30 (which roughly maps onto approximately how many trials an agent spends in a latent attention state), and

then used a non-linear transformation to compute $\tau$: 1-(1/T). The value of $\tau$, thus quantifies the probability of transitioning between the two states. The agent was initialized to be in an attentive state at the onset of trials.

The likelihood of this model can be computed:

$$
\begin{aligned}
\mathcal{L}(\theta) \quad &= \sum_{t=1}^{T} \log \mathbb{P}(a_t | h_t, \bar{h}_{t-1}, \theta) \\
&= \sum_{t=1}^{T} \log \left( \sum_{l} \mathbb{P}(a_t | h_t, ls_t = l; \theta) \mathbb{P}(ls_t = l, \bar{h}_{t-1}; \theta) \right)
\end{aligned}
$$

where $ls$ = latent state, $l \in \{$ 0 = disengaged state, 1 = engaged state $\}$, $\bar{h}_{t-1}$ corresponds to the history of actions and rewards up to the trial $t$. However, it is in practice intractable, because of the sum over latent states in the equation, which cannot be factored out.

**Cognitive models of the HRL task.** **Bayesian models of the HRL task.** Bayesian models of the HRL task assume an inference process over the latent variable of which arrow is currently the valid arrow, and thus which side (R/L) (given the current trial's set of arrows) is most likely to result in positive outcome. The inference relies on the generative model of the task determined by parameters $p_{switch}$ and $p_{reward}$, history of trial observations $O_t$, set of arrows and stochastic choice based on this inference. Initial prior belief over arrows is initialized uniformly $prior = 1/nA$, where $nA$ corresponds to the number of arrows.

To determine the agent policy over arrows at trial $t$, we first implemented a softmax function with decision parameter $\beta$ and prior belief of which arrow is the correct one; once the arrow is selected, the agent implements an $\epsilon$-greedy policy conditioned on the selected arrow $A_t$ to choose a R/L side:

$$
P(side(A_t) | A_t) = 1 - \epsilon
$$

Likelihood $p(r_t = 1 | A_t, side(A_t))$ and posterior are then updated into the prior belief for the next trial using the $p_{switch}$ model of the task parameter:

$$
p(C_{t+1} = i | O_{1:t-1}) = (1 - p_{switch}) * p(C_t = i | O_{1:t-1}) + p_{switch}(1 - p(C_t = i | O_{1:t-1}))
$$

This belief is subsequently used to inform arrow choices on the next trial. This model differs from the Bayesian Inference model for the probabilistic task in that 1) $p_{reward}$ and $p_{switch}$ parameters are not free/inferred and 2) the choice of the side is stochastic, allowing for a potential lapse in selecting the side that is not consistent with the selected arrow. This model, thus had has following free parameters: decision parameter $\beta$ and noise parameter $\epsilon$. Like in the Bayesian inference model for the PRL task, we also tested the model variant with stickiness $\kappa$ parameter that biases beliefs associated with the arrow/side chosen on the previous trial. Both models have tractable likelihoods.

**Hierarchical reinforcement learning.** We also simulated a simple hierarchical reinforcement learning (HRL) model to simulate the performance on a HRL task (S1 Fig). This model assumes that an agent tracks the value of each of the arrows, and chooses between the arrows noisily:

$$
P(arrow) = \frac{\exp(\beta \, Q_t(arrow))}{\sum_{i=1}^{n_A} \exp(\beta \, Q_t(arrow_i))} \tag{5}
$$

We have also explored the model with an assumption that an agent has a tendency to repeat the choice from the previous trial, captured by the stickiness parameter $\kappa$:

$$P(arrow) = \frac{\exp(\beta \ Q_t(arrow) + \kappa(arrow = arrow_{t-1}))}{\sum_{i=1}^{n_A} \exp(\beta \ Q_t(arrow_i))} \tag{6}$$

Once the agent chooses the arrow, it greedily chooses the direction based on which side (left/right) the arrow is pointing at (observable). Note that we only know the side the agent selected (left/right), because the arrow the agent chooses is non-observable. The agent then observes an outcome, and updates the value of the selected arrow based on the observed outcome:

$$Q_{t+1}(\text{arrow}) = Q_t(\text{arrow}) + \alpha(r - Q_t(\text{arrow}))$$

In the case of this model, the likelihood is intractable because of the need to integrate over uncertainty of what rule (which arrow) the agent followed on all of the past trials; because the integration exponentially increases with each time point, the likelihood is not tractable beyond the first several trials:

$$\begin{aligned} \mathcal{L}(\theta) &= \sum_{t=1}^{T} \log\mathbb{P}(a_t|h_t, \bar{h}_{t-1}, \theta) \\ &= \sum_{t=1}^{T} \log\mathbb{P}\left(\sum_c \mathbb{P}(a_t|h_t, rule_t = c; \theta)\mathbb{P}(rule_t = c, \bar{h}_{t-1}; \theta)\right) \end{aligned}$$

where $a_t$ corresponds to the action dictating which side the agent selected (left/right), $\bar{h}_{t-1}$ corresponds to the task history encoding rewards, selected actions/sides, arrow directions, and c correspond to identity/color of the correct arrow.

## Likelihood-dependent methods

**Maximum likelihood and maximum a posteriori estimation.**   Maximum likelihood estimation (MLE) represents a cornerstone of modeling that leverages probability theory and estimation of likelihood ($P(D|M, \theta)$) of the data given the model parameters and assumptions [18]. The parameter estimates are determined as the values that maximize the likelihood of the data:

$$\theta_{MLE} = argmax P(D|\theta)$$

$$= argmax \prod_i P(D_i|\theta)$$

$$= argmax \sum_i log P(D_i|\theta)$$

Thus, to estimate best fitting parameters via MLE, the likelihood of the data is computed and maximized with respect to parameter values via an optimization algorithm (often a blackbox one, such as fmincon in MATLAB or optimize.minimize from scipy toolbox in python). Maximum a posteriori estimation (MAP) relies on much the same principle, with an

addition of a prior $p(\theta)$ to maximize the posterior:

$$\theta_{MAP} = argmax \sum_i logP(D_i|\theta)logP(\theta)$$

As a prior for the MAP approach, we used an empirical prior derived from the true simulating distribution of parameters. We note that this gives an advantage to the MAP method above what would be available for empirical data, allowing MAP to provide a ceiling performance on the test set.

Because MAP and MLE rely on likelihood computation, their use is essentially limited to models with tractable likelihood. We used MAP and MLE to estimate parameters of tractable-likelihood models as one of the benchmarks against which we compared our ANN approach. Specifically, we fit the models to the test-set data used to compute the MSE of the ANN, and compared fit using the same metric across methods (see main text).

**Likelihood approximation methods.** Because models with tractable likelihood comprise only a small subset of all possible (and likely more plausible) models, researchers have handled the issue of likelihood intractability by implementing various likelihood approximation methods. While there are different likelihood approximation tools, such as particle filtering [31] and assumed density estimation [32], we focus on Approximate Bayesian Computation (ABC; [33, 36, 37, 70]), as it is more widely accessible and does not require more extensive mathematical expertise. ABC leverages large scale model simulation to approximate likelihood. Specifically, a large synthetic data set is simulated from a model, with parameters randomly sampled from a specific range for each agent. Summary statistics that describe the data (e.g., average accuracy or variance in accuracy) are used to construct the empirical likelihood that can be used in combination with classic methods.

We implemented a basic form of ABC—the rejection algorithm [33]. This algorithm first samples a set of model parameters $\theta$, simulates the data $\hat{D}$ from the model $M$ using these parameters and computes the predetermined summary statistic $S(\hat{D})$ of the simulated data which we refer to as the sample. The summary statistics of the real data $S(D)$ and the sample $S(\hat{D})$ are then compared—if the distance between the two sets of summary statistics $\rho$ is greater than the predetermined criterion $\epsilon$, the sample is rejected:

$$\rho(S(\hat{D}), S(D)) \leq \epsilon$$

The distance metric, like the rejection criterion, is determined by the researcher. The samples that are accepted are the samples with distance to the real data smaller than the criterion, resulting in the conclusion that parameters used to generate the sample data set can plausibly be the ones that capture the target data. Thus, the result of the ABC for each data set is a distribution of plausible parameter values which can be used to obtain point estimates via the mean, median, etc.

ABC is a valuable tool, but standard ABC has serious limitations [33]. For instance, the choice of summary statistics is not a trivial problem, and different summary statistics can yield significantly different results. Similarly, in the case of rejection algorithm ABC, researchers must choose the rejection criterion which can also affect the parameter estimates. A possible way to address this is using cross validation to determine which rejection criterion is the best, but this also requires specification of the set of possible criteria values for the cross validation algorithm to choose from. Furthermore, one of ABC assumptions is the independence of data points, which is violated in many sequential decision making models (e.g. reinforcement learning).

To compare our approach to ABC, we used network training set data as a large-scale simulation data set, and then estimated parameters of the held-out test set also used to evaluate the ANN.

To apply ABC in our case, we needed to select summary statistics that adequately describe performance on the task. We used the following summary statistics to quantify the agent for the models simulated on the PRL task:

- Learning curves: We computed agents' probability of selecting the correct action, aligned to the number of trials with reference to the reversal point. Specifically, for each agent we computed an average proportion of trials where a correct action was selected N trials before and N trials after the correct action reversal point, for all reversal points throughout the task. This summary statistic should capture learning dynamics, as the agent learns to select the correct action, and then experiences dip in accuracy once the correct actions switch, subsequently learning to adjust based on feedback after several trials.

- 3-back feedback integration: The 3-back analysis quantifies learning as well; however, instead of aligning the performance to reversal points, it allowed us to examine agents' tendency to repeat action selection from the previous trial contingent on reward history—specifically the outcome they observed on the most recent 3 trials. Higher probability of repeating the same action following more positive feedback indicates learning/sensitivity to reward as reported in [11]

- Ab-analysis: The Ab-analysis allowed us to quantify probability of selecting an action at trial $t$, contingent both on previous reward and action selection history (trials $t - 2$ and $t - 1$, [11, 80]).

For the models simulated on a hierarchical task we used the learning curves as summary statistics (same as for the PRL), where reversal points were defined as the switch of the correct rule/arrow to follow. In addition, we quantified the agent's propensity to stick with the previously correct rule/arrow, where the agent should be increasingly less likely to select the side consistent with the arrow that was correct before the switch, as the number of trials since the switch increases. Similarly, we used a version of the 3-back analysis where the probability of staying contingent on the reward history referred to the probability of potentially selecting the same cue across the trial window, based on observed choices of the agent. All summary statistics are visualized in S8 Fig.

**Model comparison.** To perform benchmark model comparison, we used the Akaike Information Criterion (AIC) metric [5], commonly used to evaluate relative model fitness, with the aim of identifying the best model candidate that might have generated the data. The AIC score combines model log likelihood and number of parameters to quantify model fitness, while also penalizing for model complexity in order to prevent overfitting:

$$AIC = -2(LLH) + 2K$$

where $K$ corresponds to the number of parameters. The model with the lowest AIC scores corresponds to the best fitting model for the given data. A related metric that is commonly used is the Bayesian Information Criterion (BIC, [22]), which considers the number of observations ($N$) as well, and similarly uses the lowest score to signal the best fitting model:

$$BIC = -2(LLH) + K*log(N)$$

We used AIC score as it outperformed BIC model comparison, and thus provided us with ceiling benchmark to evaluate the ANN.

To perform proper model comparison, it is essential to not only evaluate the model fitness (overall AIC/BIC score), but also to test how reliably the true models (that generated the data) can be identified and successfully distinguished from others. To do so, we constructed a confusion matrix based on the AIC score (Fig 5A). We used the test set data simulated from each model, and then fit all candidate models to each of the data sets while also computing the AIC score for each fit. If the models are identifiable, we should observe that AIC scores for true models (e.g., the models the data was simulated from) should be the lowest for that model when it's fit to the data compared to other model candidates.

## Artificial neural network-based method

**Parameter recovery.**   To implement ANNs for parameter estimation we have used the relatively simple neural network structure inspired by the previous work [52]. In all experiments, we used 1 recurrent GRU layer followed by 3 fully connected dense layers with 2000 dimensional input embeddings (S1 Table). To train the network, we simulated a training data set using known parameters. For each model, we used 30000 training samples, 3000 validation samples, and 3000 test samples that are generated from simulations separately. For probabilistic RL, the input sequence consisted of rewards and actions. For hierarchical RL, the sides (left/right) of three arrow stimuli are added to the rewards and actions sequences. The network output dimension was proportional to the number of model parameters. We used a *tanh* activation in the GRU layer, *ReLu* activations in 2 dense layers, and a linear activation at the final output. Additional training details are given below:

- We used *He* normal initialization to initialize GRU parameters [81].

- We used the Adam optimizer with mean square error (MSE) loss and a fixed learning rate of 0.003. Early stopping (e.g. network training was terminated if validation loss failed to decrease after 10 epochs) was applied with a maximum of 200 epochs.

- We selected network hyperparameters with Bayesian optimization algorithms [82] applied on a validation set. Details of the selected values are shown in S1 Table.

All of the training/validation was run using TensorFlow [83]. The training was performed on Nvidia V100 GPUs with 25 GB memory.

**Network evaluation.**   The network predicted the values of parameter on the test set that is unseen in the training and validation. We also conducted robustness tests by varying trial numbers (input size).

To evaluate the output of both ANN and traditional tools we used the following metrics (ensuring our results are robust to the choice of performance quantification):

- The mean squared error (MSE): To evaluate parameter estimation accuracy, we calculated a mean squared error between true and estimated model parameter across all agents. Prior to calculating MSE all parameters were normalized, to ensure comparable contribution to MSE across all parameters. Overall loss for a cognitive model (across all parameters) was an average of individual parameter MSE scores. Overall loss for a class of models (e.g., likelihood-tractable models) was an average across all model MSE scores.

- Spearman correlation ($\rho$): We used Spearman correlation as an additional metric for examining how estimated parameter values relate to true parameter values, with higher Spearman $\rho$ values indicating higher accuracy. We paired Spearman correlations with scatter plots, to visualize patterns in parameter recoverability (e.g., whether there are specific parameter ranges where parameters are more/less recoverable).

- R-Squared ($R^2$ or the coefficient of determination): R-Squared represents the proportion of variance in true parameters that can be explained by a linear regression between true and predicted values. It thus indicates the goodness of fit of an ANN model. We calculated an R-Squared score for each individual parameters across all agents and used it as an additional evaluation for how well the data fit the regression model.

**Uncertainty estimation.**    To compute uncertainty of parameter estimates we have incorporated evidential learning into our method [58]. In the application of evidential learning to continuous regression [58] observed targets follow a Gaussian distribution, characterized by its mean and variance. Conjugate Gaussian prior, normal inverse-gamma distribution, is created by placing a prior on both the mean (Gaussian distribution) and the variance (inverse-gamma distribution). By sampling from this distribution, specific instance of the likelihood function is obtained (based on both mean and the variance). This approach not only aims for accurate target predictions but also takes into account the uncertainty (quantified by the variance term). For more insights and details into evidential learning, refer to the work by [58]. Their research also introduces a regularization term, which is useful for penalizing incorrect evidence and data that falls outside the expected distribution.

For the purpose of visualization (Fig 4), we have created upper and lower bounds of targets by adding/subtracting variance from the predicted target values. We then re-scaled these values by applying the inverse scaler (e.g., from the scaler applied to normalize parameters for network training). This provides a scale-appropriate and more interpretable visualization of parameter recovery and uncertainty for each parameter.

**Alternative models.**    We have also tested the network with long short-term memory (LSTM) units since LSTM units are more complex and expressive than GRU units; nevertheless they achieved the similar performance as GRU units but are more computationally expensive, and thus we mostly focused on the GRU version of the model. Since LSTM worked, but not better than GRU, the LSTM results are reported in S2–S5 Figs.

**Model identification.**    The network structure and training process were similar to that of the network used for parameter recovery, with an exception of the output layer that utilized categorical cross-entropy loss and a softmax activation function. The network validation approach was the same as the one we used for parameter recovery (e.g., based on the held-out test set). We also observed a better performance when training with various trial numbers.

**Robustness test: Influence of different input trial numbers.**    For all robustness experiments, we followed the same training procedures as described previously while varying the training data. The details of training data generation are given below:

**Parameter recovery.**    We simulated 30,000 training samples with 2000 trials per simulation in the probabilistic reversal learning task. For shorter fixed trial sequence lengths per training samples (e.g., 500), we used the same training set truncated to the first 500 trials. To generate the training data with different trial numbers across training samples, we reused the same training set, with sequences of trials truncated to a given number. There were 6000 training samples of 50, 100, 500, 1000, 1500, and 2000 trials, each.

**Model identification.**    The process of data generation for model identification robustness checks was similar to parameter recovery. However, we only simulated 500 trials for each model because we found no significant increase in accuracy with higher trial numbers.

## Supporting information

**S1 Fig. Tasks.** A) Probabilistic Reversal Learning task. We simulated artificial agents using cognitive models of behavior on a Probabilistic reversal learning (PRL) task, which provides a

dynamic context for studying reward-driven learning. In this task, an agent chooses between two actions, where one of the actions gets rewarded with higher probability ($p(r)$ =.80) and one with lower ($1 - p$). After a certain number of correct trials, the reward probabilities of the two actions reverse. The task provides an opportunity to observe how agents update their model of the task (e.g. correct actions) based on observed feedback. B) Hierarchical reinforcement learning task. In this task, three differently colored arrows represent three potential rules an agent can follow when selecting one of the two actions (left/right) corresponding to the side the chosen arrow is pointing at. Selecting a side consistent with correct arrow is rewarded with probability $p$ =.90. Correct arrow switches after a certain number of trials. The task provides a possibility to examine how following latent rules may shape agents' learning behavior.
(TIF)

**S2 Fig. 2 Parameter RL (2P-RL) model parameter recovery using different fitting methods.** $\rho$ corresponds to Spearman correlation coefficient. The red line represents a unity line (x = y), and the black line represents a least squares regression line.
(TIF)

**S3 Fig. 4 Parameter RL model (4P-RL) parameter recovery using different fitting methods.** $\rho$ corresponds to Spearman correlation coefficient. The red line represents a unity line (x = y), and the black line represents a least squares regression line.
(TIF)

**S4 Fig. Bayesian Inference (BI) model parameter recovery using different fitting methods.** $\rho$ corresponds to Spearman correlation coefficient. The red line represents a unity line (x = y), and the black line represents a least squares regression line.
(TIF)

**S5 Fig. Bayesian Inference with stickiness (S-BI) model parameter recovery using different fitting methods.** $\rho$ corresponds to Spearman correlation coefficient. The red line represents a unity line (x = y), and the black line represents a least squares regression line.
(TIF)

**S6 Fig. Bayesian Inference with stickiness (S-BI) model parameter recovery using different fitting methods.** Correlation between the average experienced time intervals in the attentive state and the $\tau$ parameter in RL-LAS model that captures transition between disengaged/engaged attention states estimated by the ANN.
(TIF)

**S7 Fig. Misclassification of Bayes and sticky Bayes model.** Misclassification of Bayes and sticky Bayes model is contingent on the value of the stickiness parameter $\kappa$. The misclassification percentage is higher at $\kappa$ values closer to 0.
(TIF)

**S8 Fig. Summary statistics for Approximate Bayesian Computation (ABC).** Top row shows summary statistics computed for all models simulated on a probabilistic reversal learning task; the figure only shows agents simulated using a 4-parameter RL model. The bottom row shows summary statistics computed for all models simulated on a hierarchical reversal learning task; the figure only shows performance of HRL model agents. Both rows depict 200 out of 3000 test set agents. Gray lines represent individual agents, and the black line represents an average across the agents.
(TIF)

**S9 Fig. Effect of prior misspecification on parameter estimation in MAP and our ANN approach.** A) Applying too narrow a prior specification to the fitting procedure (prior in MAP, training samples in ANN) results in difficulty estimating out-of-range parameters for both MAP and ANN. Broader prior specification addresses this issue, with only a slight loss of precision in specific target ranges. Training the network with a broad range of parameters while oversampling parameters from regions of interest yields the most robust results. B) Visualization of fitting with MAP and ANN with a wide prior, tested on a full range/wide range data set—training the network with broader range while oversampling from the most plausible range yields less noisy performance in the range compared to MAP. Red lines delineate the range of the narrow prior, which corresponds to the main text results. C) The broad prior was designed by sampling from the full broader range ($\beta \in [0, 10]$, $\alpha \in [0, 1]$), with the constraint that 70% of samples are in the expected narrow range ($\beta \in [2, 6]$, $\alpha \in [0.5, 1]$, and 30% outside).
(TIF)

**S10 Fig. Consistency between methods for parameter estimation.** A) The correlation between parameter estimates in the 4P-RL model derived using MAP and ANN is high, and indeed stronger than the correlation between true and derived parameters (see Fig 2). This indicates that both methods systematically misidentify some parameters similarly, likely due to specific data patterns. B) The correlation between parameter estimates in the Bayesian inference model derived using MAP and ANN shows similar results.
(TIF)

**S11 Fig. Consistency between methods for parameter estimation in two model misspecification cases.** A) The correlation between MAP and GRU RL model parameter estimates, fit to data simulated from Bayesian Inference model. B) The correlation between MAP and GRU Bayes model parameter estimates, fit to data simulated from the RL model. High correlation would imply similarities in estimates between MAP and GRU, suggesting that ANNs are similarly impacted by model misspecification as traditional methods such as MAP.
(TIF)

**S12 Fig. Comparison of model predictions of ground truth simulated behavior (black line) and choices simulated using A) MAP and B) GRU estimated parameters (gray line) of the 4P-RL model.** We randomly sampled 100 agents from the test set, and the respective parameter estimates for each of the methods. We simulated data from the model and compared it to ground truth. Both methods successfully recover choices from the ground truth agents.
(TIF)

**S13 Fig. Effect of model misspecification on standard method and ANN performance.** We randomly sampled 100 agents from the test set, and the respective parameter estimates for each of the methods. We then simulated data from the model and compared it to ground truth. Both methods successfully recover choices from the ground truth agents. A) We fit the Bayesian Inference model (without stickiness) to the data simulated from the Bayesian Inference Model with stickiness using MAP. We correlated the estimated Bayesian inference model parameters (y-axis) with the ground truth parameters from the model with stickiness (x-axis). B) For the ANN, we trained the neural network to estimate parameters of the Bayesian inference model, and tested it on the data simulated from the Bayesian inference model with stickiness. We looked at the correlation between the ground truth parameters (from a separate test set), and the predictions of the network trained on the model without stickiness. Both methods show that parameters shared between the misspecified models can be reasonably and similarly recovered. Both ANN and MAP generated some non-zero estimates of stickiness when data

simulated from model without stickiness was fit using the model/network that assumes presence of stickiness in the model; however, these values were closely clustered around 0, to a similar degree between methods (S16 Fig).
(TIF)

**S14 Fig. Effect of model class misspecification on standard method and ANN performance.** We fit the RL model to the data simulated from the Bayesian Inference model in the probabilistic reversal learning task (see Methods section on Tasks and Cognitive Models) using A) MAP and B) GRU. We correlated the estimated RL model parameters (y-axis) with the ground truth parameters from the Bayesian inference model (x-axis). MAP and GRU show similar patterns between estimated and true parameters, such that variance driven by true parameter $\beta$ $p_{switch}$ are both captured in the fit $\beta$ parameters, while the fit learning rate parameter $\alpha$ captures behavioral variance driven by the Bayesian update parameters $p_{reward}$ and $p_{switch}$.
(TIF)

**S15 Fig. Effect of model class misspecification on standard method and ANN performance.** We fit the Bayesian Inference model to the data simulated from the RL model in the probabilistic reversal learning task (see Methods section on Tasks and Cognitive Models) using A) MAP and B) GRU. We correlated the estimated Bayesian inference model parameters (y-axis) with the ground truth parameters from the RL model (x-axis). MAP and GRU again show similar patterns between estimated and true parameters. In particular, we see that in both cases, noise in behavior due to $\beta$ in the RL model tends to be attributed to the $p_{switch}$ fit parameter rather than the fit Bayesian model $\beta$ parameter. Effect of learning rate parameter $\alpha$ are attributed to $p_{reward}$ by both methods.
(TIF)

**S16 Fig. Stickiness parameter estimates.** Stickiness parameter estimates from the data simulated from the Bayesian inference model without stickiness from A) fitting the Bayesian inference model with stickiness using MAP, and B) utilizing the ANN trained to estimate parameters of the model with stickiness. Despite both methods producing non-zero estimates of stickiness, they tend to cluster around the value of 0.
(TIF)

**S17 Fig. Neural network performance variability by different seeds for model identification and parameter estimation.** For conciseness, we show tests from 10 different seeds on model identification with 4 models simulated on the PRL task (e.g. same as Fig 5) and parameter estimation of one of the likelihood-intractable models (e.g. RL-LAS model). We found that overall both model identification and parameter estimation had relatively stable results across different seeds, with an exception of one seed value in both cases.
(TIF)

**S1 Table. Summary of hyper-parameter values selected from Bayesian optimization algorithms.**
(XLSX)

## Acknowledgments

We thank Jasmine Collins, Kshitiz Gupta, Yi Liu and Jaeyoung Park for their contributions to the project. We thank Bill Thompson and all CCN lab members for useful feedback on the project.

## Author Contributions

**Conceptualization:** Milena Rmus, Liyu Xia, Anne G. E. Collins.

**Data curation:** Milena Rmus, Ti-Fen Pan.

**Formal analysis:** Milena Rmus, Ti-Fen Pan.

**Funding acquisition:** Anne G. E. Collins.

**Investigation:** Milena Rmus, Liyu Xia, Anne G. E. Collins.

**Methodology:** Milena Rmus, Ti-Fen Pan, Liyu Xia.

**Project administration:** Anne G. E. Collins.

**Software:** Milena Rmus, Ti-Fen Pan, Liyu Xia.

**Supervision:** Anne G. E. Collins.

**Validation:** Milena Rmus, Ti-Fen Pan.

**Visualization:** Milena Rmus, Ti-Fen Pan.

**Writing – original draft:** Milena Rmus, Ti-Fen Pan, Liyu Xia, Anne G. E. Collins.

**Writing – review & editing:** Milena Rmus, Ti-Fen Pan, Liyu Xia, Anne G. E. Collins.

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
