## [Decision Letter · Decision Letter 0]

7 Nov 2023

Dear Ms Rmus,

Thank you very much for submitting your manuscript "Artificial neural networks for model identification and parameter estimation in computational cognitive models" for consideration at PLOS Computational Biology.

As with all papers reviewed by the journal, your manuscript was reviewed by members of the editorial board and by several independent reviewers. In light of the reviews (below this email), we would like to invite the resubmission of a significantly-revised version that takes into account the reviewers' comments.

Both Reviewers 1 and 2 acknowledged the importance of the work for computational cognitive modeling, while pointing out shortcomings that, if addressed, may significantly strengthen the paper. Please address these suggestions as best as you can. Reviewer 3 pointed out a disconnection of the current manuscript from the literature of amortized Bayesian inference. This is also reflected in the comments of the other reviewers. I believe the disconnection is not intentional but rather reflects a slow penetration of the new development in this field to psychology/neuroscience. Nonetheless, it is necessary to make a better connection to the literature suggested by the reviewers and explain the contribution more fairly and clearly, especially for the target domain of application.

We cannot make any decision about publication until we have seen the revised manuscript and your response to the reviewers' comments. Your revised manuscript is also likely to be sent to reviewers for further evaluation.

Sincerely,

Ming Bo Cai

Academic Editor

PLOS Computational Biology

Daniele Marinazzo

Section Editor

PLOS Computational Biology

Both Reviewers 1 and 2 acknowledged the importance of the work for computational cognitive modeling, while pointing out shortcomings that, if addressed, may significantly strengthen the paper. Please address these suggestions as best as you can. Reviewer 3 pointed out a disconnection of the current manuscript from the literature of amortized Bayesian inference. This is also reflected in the comments of the other reviewers. I believe the disconnection is not intentional but rather reflects a slow penetration of the new development in this field to psychology/neuroscience. Nonetheless, it is necessary to make a better connection to the literature suggested by the reviewers and explain the contribution more fairly and clearly, especially for the target domain of application.

Reviewer's Responses to Questions

**Comments to the Authors:**

Reviewer #1: In this manuscript, the authors suggest a simulation-based inference

technique to make inferences about the behavioral model and parameters

within such a model to fit (in this case behavioral) data. They do this

by training a recurrent neural network (RNN) to ingest choices and

report appropriate parameters or the posterior over the model

identity. Training is based on simulating behavior using a suitable

range of parameters and models, and performing supervised learning. The

authors show that this works very well in cases in which likelihoods are

available (often outperforming traditional likelihood-based methods) and

when they are not (often outperforming an approximate Bayesian

computation or ABC method). Some tests of robustness are provided.

The manuscript makes a clear and compelling contribution - and offers a

method that, if it can be shown to work on the sort of out-of-sample

data that humans and animals generate, could become an important weapon

in the modeling armory.

Some comments:

- the manuscript does not really do justice to the ever-growing

collection of simulation based inference methods. Although it cites

the Boelts et al paper that uses a neural network to generate a

likelihood in cases in which likelihoods do not exist, it does not

cite or compare itself to other methods such as Lueckmann, Goncalves

et al, NeurIPS (2017) which also learn a direct map from sample data,

but in this case to posterior distributions over parameters (and not

for behavioral data). This seems like a stronger point of comparison -

since it is often most useful to know the full posterior.

Similarly, it is not the case that it is always necessary to provide

ABC with summary statistics - there are learning methods for

this.

The review paper on 'Simulation Intelligence' by Lavin et al (2022)

has quite an extensive survey.

- if we understand it correctly, the method for model comparison seems

pretty unwieldy, since it is trained discriminatively, needing to be

completely retrained every time an additional model is considered. An

approximate likelihood method would have a big advantage here.

- that MAP estimation performs worse than the RNN in the case that the

likelihood is tractable (and the prior is correct; e.g., fig 2) makes

one worry that the optimization scheme employed is inefficient.

- for practical use, it would be important to understand a bit more

about things like the impact of the size of the RNN or other parameter

choices - particularly since training recurrent nets has its

difficulties.

- the tests of robustness were a little underwhelming. The effect of

partial model mis-specification on parameter estimation (e.g., train

with no lapse parameter; fit on data with a lapse parameter) would be

interesting.

- Uncertainty, as delivered by fully Bayesian approaches, can serve to

identify problems with parameters trading-off. It might be worth

discussing the possibility of uncertainty estimates through this method.

- In Fig 2a, the mistakes the RNN makes for alpha^+ seem quite systematic,

could a word be said about why the model cannot overcome these errors?

- Minor points:

- There are some articles missing, e.g. line 112: "across the respective",

line 118: "While the fitting process", line 120: "models, the ANN

performed" or "models, ANNs performed", and a few more throughout the

text.

- The unity line is y=x, not x=x.

- In line 102 the abbrevations MLE and MAP are given in the wrong order.

- LLH is used in Figure 1 but never spelled out as log-likelihood.

- Line 191 and elsewhere, we think it should be the number of trials, not

length.

- The caption in Figure 5 mention RL-LAS and 2P-RL, possibly the latter

should be HRL?

Reviewer #2: Computational cognitive models of behavior play key roles in psychology and neuroscience. They are at their most powerful when they provide quantitative predictions about behavioral and neural data, which requires fitting their parameters to a dataset. Standard ways of doing this involve computing the likelihood function p(dataset | model, parameters). However, interesting cognitive models exist for which this function is intractable to compute, making them difficult to use. This paper explored a novel and creative solution to this problem: training artificial neural networks on the key tasks of parameter estimation and model comparison. For parameter estimation, this involves generating a large number of synthetic datasets using a single model with known generative parameters, and training an ANN on the mapping between dataset and parameters. For model selection it involves generating synthetic datasets from a variety of different known models, and training an ANN on the mapping from dataset to model identity. Authors show that both methods work well on synthetic datasets.

Overall I think this is a great paper. The problem addressed is an important one. The approach is creative and an interesting new direction for the field. I do think the manuscript could do more to discuss possible limitations which seem likely to arise in moving from synthetic datasets generated by artificial agents to the intended application of laboratory datasets generated by humans or animals. I also think it could do more to discuss pros and cons of the approach relative to a broader range of previously-published alternative approaches.

Major comments

1. A known issue with ANNs is that they can behave incorrectly and unpredictably when given inputs outside the range of their training distribution. In this paper’s approach, the training distribution consists of behavior datasets generated by one or more of the cognitive models being considered. In general, it seems to me that we should expect laboratory datasets to be at least somewhat outside of this distribution, even for the very best cognitive models (“all models are wrong”), raising the possibility that ANN-based parameter estimates might be incorrect or inconsistent. Note that parameter estimates from wrong models can still be scientifically useful, for example in characterizing behavioral differences between experimental groups (and also that even our very best models are almost certainly at least slightly wrong).

A revised manuscript might explore the extent of this issue further by performing parameter estimates for models other than the generative model (either using laboratory datasets or mis-matched synthetic datasets). Are these parameter estimates consistent within a dataset between different ANN random seeds? For models with tractable likelihoods, are they consistent with the parameter estimates produced by classic methods? Alternatively, if authors consider this exploration out of scope, a revised manuscript should at least contain a careful discussion of these issues.

2. The introduction lists a number of alternative approaches to parameter estimation using models with intractable likelihood functions: Approximate Bayesian Computation, Inverse Bayesian Sampling, particle filtering, assumed density estimation, and likelihood approximation networks. Of these, only likelihood approximation networks and ABC are engaged with in detail in the discussion section. It would strengthen the paper to briefly engage with each of the other methods, explaining pros and cons of the ANN-based approach.

In particular, the Inverse Bayesian Sampling approach seems to me to have many of the advantages cited for the ANN-based approach: it allows treating the cognitive model as a black box from which behavioral data are sampled, and does not require the user to think in any detail about the model’s inner workings. IBS may have advantages over the ANN approach (it likely does not suffer from the training-distribution issue described above), and likely has disadvantages as well (perhaps it is computationally inefficient; certainly its computational demands scale with the number of subjects, while the ANN can perform its expensive computations once then fit new subjects very cheaply; etc). It would be helpful for the manuscript to include the authors’ thoughts on this and on other points of comparison between their method and previously-published methods for parameter estimation in models with intractable likelihoods.

3. Model selection raises a number of issues, many of which are not unique to the ANN-based approach, but which should be discussed in its context to assist readers. These issues are particularly acute when it comes to selection among nested models. It seems likely to me that the ANN-based approach shares a key limitation with the classic Bayes Factor approach (e.g. via the Savage-Dickey method), in that model selection results will be highly sensitive to the prior over model parameters. This is a major limitation because there is rarely a scientifically principled way to select such a distribution, and because changes in the distribution can easily swing a model comparison from “very strongly favors the simpler model” to “very strongly favors the more complex model”.

It seems to me that the ANN approach is likely to share this limitation, with the prior over parameters expressed as the generative distribution of parameters used to create the ANN’s training dataset. If the authors agree, I think this should be briefly discussed. Regardless, it would be helpful to readers to discuss briefly what kinds of scientific conclusions can and cannot safely be drawn from an ANN-driven model selection. Some of these limitations will be shared with classic methods: for example it is always a worry that the “true” model might not be among the set of models being compared – this worry might be particularly acute here if the dataset puts the ANN outside its training distribution.

Minor comments

1. “... for most models, [researchers can] simulate synthetic data to make predictions”. Surely this is the case for all models? In what sense is something a “model” if it can’t be used to simulate data and make predictions?

2. Figure 1: What is “LLH”? I do not think this is defined anywhere

3. Estimating quality of parameter estimation using parameter-by-parameter MSE is a nonstandard choice and should be discussed. It raises the possibility that parameter estimation accuracy will depend critically on model parameterization (for example it will change if parameters are rescaled, or reparameterized to orthogonalize them). It is also likely the explanation for the otherwise-paradoxical finding that ANNs outperform MAP, which is a Bayes optimal way to estimate parameters. For some scientific applications, like comparing parameter estimates between experimental groups, there might be a case to be made that MSE is actually better than standard approaches. For others it is likely worse (though perhaps negligibly so) or equivalent. These things should be discussed for the benefit of the reader.

4. Figure 2A: Unclear if errorbars reflect variability over “participants” (for a fixed random seed), over network seeds (for a fixed dataset), or something else. Variability over seeds is something that should be clearly shown. If you were to repeat your complete procedure, which I believe includes fitting multiple models in order to select hyperparameters, many times, how much variability in final results would there be?

Likewise for Figure 2B. Is this an example network? If so, make that explicit and show something about the variability among networks. And likewise for the remaining figures. Figures 4 and 5 could show standard deviations over seeds.

<br

---

## [Decision Letter · Decision Letter 1]

7 Mar 2024

Dear Ms Rmus,

Thank you very much for submitting your manuscript "Artificial neural networks for model identification and parameter estimation in computational cognitive models" for consideration at PLOS Computational Biology.

As with all papers reviewed by the journal, your manuscript was reviewed by members of the editorial board and by several independent reviewers. In light of the reviews (below this email), we would like to invite the resubmission of a significantly-revised version that takes into account the reviewers' comments.

Both Reviewers 1 and 3 expressed concerns with not performing an evaluation or implementation of posterior estimation rather than just point estimate. Therefore, it seems necessary to include this demonstration in the paper. Other comments from Reviewers 3 are also important and should be properly addressed. In addition, although not all of the researchers in cognitive modeling are aware of the recent progress in simulation-based inference, given what reviewers have pointed out, and some list of the existing works: https://github.com/smsharma/awesome-neural-sbi, it would be great to not over-state the contribution of the work and provide proper acknowledge of the existing contribution. Demonstrating the method on real data as suggested by Reviewer 2 especially from the domain in which the authors are familiar with would be a good idea to further strengthen the paper.

We cannot make any decision about publication until we have seen the revised manuscript and your response to the reviewers' comments. Your revised manuscript is also likely to be sent to reviewers for further evaluation.

Sincerely,

Ming Bo Cai

Academic Editor

PLOS Computational Biology

Daniele Marinazzo

Section Editor

PLOS Computational Biology

Both Reviewers 1 and 3 expressed concerns with not performing an evaluation or implementation of posterior estimation rather than just point estimate. Therefore, it seems necessary to include this demonstration in the paper. Other comments from Reviewers 3 are also important and should be properly addressed. In addition, although not all of the researchers in cognitive modeling are aware of the recent progress in simulation-based inference, given what reviewers have pointed out, and some list of the existing works: https://github.com/smsharma/awesome-neural-sbi, it would be great to not over-state the contribution of the work and provide proper acknowledge of the existing contribution. Demonstrating the method on real data as suggested by Reviewer 2 especially from the domain in which the authors are familiar with would be a good idea to further strengthen the paper.

Reviewer's Responses to Questions

**Comments to the Authors: **

Reviewer #1: The authors have responded well to many of the points of the reviewers. Although the paper does now refer more comprehensively to the wider simulation-based inference literature, it's a little underwhelming not to have implemented/tested alternatives from this stable as a clear alternative (giving extra features such as posteriors rather than point estimates). Nevertheless, the paper's worthwhile contribution is now appropriately hedged.

Minor points:

L275: "promise [in successfully arbitrating between competing cognitive models] [in model identification]" The content of only one pair of brackets was probably meant to be in that sentence.

L278 "any standard cognitive data set with [a] normal number of participants "

Figure 3B panel ABC, True T: put "rho=.5" annotation lower

Fig S3: The top right plot changed quite a bit (and is now terrible), maybe the authors just want to double-check this is correct?

Fig S6: the p-value should be spelled out as something like "p<1e-xx"

Reviewer #2: Thank you for your thorough response to my and the other reviewers’ comments. The response fully addresses all of my concerns other than major comment #1, which it seems to misunderstand. I’ve tried to rephrase this comment below to unpack it a bit more. I’ve also described an analysis which represents one simple way of addressing it which I believe would be very straightforward to do. If the authors prefer some other way, including merely acknowledging the concern in the text, that’s of course fine too. 

Major Comment 

One major claim of this manuscript is that ANN-based parameter estimation is a reasonable substitute for MAP-based parameter estimation, and that this is a practical tool that might be useful for fitting models to real datasets generated by humans or animals. The manuscript does a great job of showing that this is true when estimating parameters from synthetic datasets where the generative process exactly matches the model being fit. But real data are never generated from exactly the model being fit (they’re generated by a human or an animal), so it’s important to check whether the ANN-based method returns similar results to MAP when applied to datasets generated by different models.

For concreteness: one way to address this would be creating a plot like the new Figure S10, which compares parameter estimates produced by MAP and by the ANN, but using mis-specified models. For example panel A showing estimates of RL model parameters would analyze datasets produced by the Bayesian model, and panel B vice-versa. Or of course the revision might choose to address it in some other way.

Reviewer #3: Attached.

**Have the authors made all data and (if applicable) computational code underlying the findings in their manuscript fully available?**

Reviewer #1: Yes

Reviewer #2: None

Reviewer #3: None

PLOS authors have the option to publish the peer review history of their article (what does this mean?). If published, this will include your full peer review and any attached files.

Reviewer #1: No

Reviewer #2: No

Reviewer #3: **Yes: **Stefan T. Radev
---

## [Decision Letter · Decision Letter 2]

27 Apr 2024

Dear Ms Rmus,

We are pleased to inform you that your manuscript 'Artificial neural networks for model identification and parameter estimation in computational cognitive models' has been provisionally accepted for publication in PLOS Computational Biology.

Best regards,

Ming Bo Cai

Academic Editor

PLOS Computational Biology

Daniele Marinazzo

Section Editor

PLOS Computational Biology

Reviewer's Responses to Questions

**Comments to the Authors:**

Reviewer #1: Although there remain some differences of interpretation between the reviewers and authors - I think the paper is fine for publication.

Reviewer #2: Thank very much for your thorough response! It has fully addressed all of my comments on the manuscript.

Reviewer #3: I appreciate that the authors have addressed my technical / detailed questions. Overall, I would have wished for a less defensive approach to my suggestions given the mounting evidence for a lack of methodological novelty in a paper initially intended to present innovative methods. Simply stating that an issue is merely a terminological one is to say that presenting Ridge regression as a novel approach despite the existence of fully Bayesian regression(s) boils down to a semantic problem.

**Have the authors made all data and (if applicable) computational code underlying the findings in their manuscript fully available?**

Reviewer #1: Yes

Reviewer #2: None

Reviewer #3: Yes

PLOS authors have the option to publish the peer review history of their article (what does this mean?). If published, this will include your full peer review and any attached files.

Reviewer #1: No

Reviewer #2: No

Reviewer #3: No

---

## [Editor Report · Acceptance letter]

9 May 2024

PCOMPBIOL-D-23-01514R2 

Artificial neural networks for model identification and parameter estimation in computational cognitive models

Dear Dr Rmus,

I am pleased to inform you that your manuscript has been formally accepted for publication in PLOS Computational Biology. Your manuscript is now with our production department and you will be notified of the publication date in due course.

With kind regards,

Zsofia Freund
